

# On the global relationship between polarimetric radio occultation observable $\Delta\Phi$ and ice water content

Ramon Padullés[1,2], Estel Cardellach[1,2], and F. Joseph Turk[3]

[1]Institut de Ciències de l'Espai, Consejo Superior de Investigaciones Científicas (ICE-CSIC), Barcelona, Spain
[2]Institut d'Estudis Espacials de Catalunya (IEEC), Barcelona, Spain
[3]Jet Propulsion Laboratory, California Institute of Technology, Pasadena, CA, USA

**Correspondence:** Ramon Padullés (padulles@ice.csic.es)

**Abstract.** The Radio Occultations and Heavy Precipitation (ROHP) experiment aboard the Spanish PAZ satellite was deployed in 2018 with the objective of demonstrating the ability of Polarimetric Radio Occultation measurement (PRO) concept in detecting rain (liquid-phase precipitation). Analysis of these data since that time have also demonstrated the ability of PRO for detection of horizontally oriented frozen-phase precipitation. To verify these observations, a global climatological comparison is performed using the Cloudsat (94-GHz) radar as a reference, for different heights and taking into account the radio occultation (limb-based) viewing geometry. A robust relationship (e.g. high correlation coefficient) is found between the polarimetric radio occultation observable $\Delta\Phi$ and the integrated ice water content along the rays for heights above the freezing level. The relationship is specially good over ocean, where the major precipitation features and the signatures of the Inter-Tropical Convergence Zone are well captured by the PAZ observations. Differences between over ocean and over land point towards different characteristics of the convective clouds above the freezing level, involving the orientation of frozen particles. The ratios between $\Delta\Phi$ and along-ray integrated ice water content are further validated with single particle forward scattering simulations, and the results indicate that a combination of horizontally oriented aggregated ice particles and tilted pristine ice plates agree well with the observations. Overall, the global climatological results obtained in this study show the presence of horizontally oriented particles across the whole globe, and for a wide range of height layers. Furthermore, the relationship between $\Delta\Phi$ and along-ray integrated ice water content can be used to attempt an inversion of the polarimetric radio occultation observations towards a potential retrieval of such water content.

## 1 Introduction

The National Academies' 2017 Decadal Survey for Earth Science and Applications from Space (National Academies of Sciences Engineering and Medicine, 2018) recommended an observing system that addresses the Aerosols, Clouds, Convection, and Precipitation (ACCP) combined designated observable, currently being formulated as the National Aeronautics and Space Administration (NASA) Atmosphere Observing System (AOS, part of the NASA Earth System Observatory) for later in this decade. The observations from the AOS instruments will be complemented by a program of record (POR) for geophysical variables (GV) that are not available at the time of AOS, including ice water path measurements being envisioned as part of the primary GV of interest related to ice-phase clouds and precipitation.





A challenging GV to measure is the precipitation and ice profile properties directly inside, and to lower levels of convective clouds, where observations from radars that operate at attenuating wavelengths are compromised owing to multiple scattering and severe attenuation (Battaglia et al., 2020). While the convective cloud vertical column is driven largely by the intensity and profile of moist convective up- and down-drafts, the convective vertical motions are controlled by exchanges between the local and regional thermodynamic environments in which convective clouds evolve (Schiro et al., 2018, 2020; Storer and Posselt,
2019). A joint profiling observation that provides an indication of convective strength alongside the thermodynamic profile may provide an observational constraint, such as representing the effects of entrainment of subsaturated air from the lower free troposphere to correctly capture the onset of convection.

The Radio Occultations and Heavy Precipitation (ROHP) experiment aboard the Spanish PAZ satellite (Cardellach et al., 2014) has demonstrated the ability of the polarimetric Radio Occultation (PRO) technique to complement well-established
RO processing techniques (e.g. Kursinski et al., 1997) with a joint detection of heavy precipitation (Cardellach et al., 2019; Padullés et al., 2022). The detection is based on the total differential phase shift ($\Delta\Phi$) accumulated along the RO ray paths, traveling from the Global Positioning System (GPS) satellite emitter to the dual linearly polarized (H, V) receiver on the PAZ Low Earth Orbiter (LEO). The $\Delta\Phi$ is measured in mm and is obtained by comparing the electromagnetic phase at both antenna ports (e.g. $\Delta\Phi = \Phi_H - \Phi_V$), assuming that the phase measurements have been previously translated to optical length by the
factor $\frac{\lambda}{2\pi}$. This differential polarimetric phase measurement, in units of length, is set as the primary observable for PRO. It depends on the hydrometeors encountered along the rays as follows (Cardellach et al., 2014):

$$\Delta\Phi = \int_L K_{\mathrm{dp}} \mathrm{d}L \qquad (1)$$

where $K_{\mathrm{dp}}$ is the differential phase shift (mm/km) and L is the traveled distance (in km). The $K_{\mathrm{dp}}$ is defined as:

$$K_{\mathrm{dp}} = 10^3 \frac{\lambda^2}{2\pi} \int_D \Re\{S_{hh} - S_{vv}\} N(D) \mathrm{d}D \qquad (2)$$

where $\lambda$ is the wavelength (i.e. in the case of GPS signals, $\lambda = 0.192$ m), $S_{hh,vv}$ are the co-polar components of the scattering amplitude matrix (m), and $N(D)$ is the particle size distribution ($\mathrm{m}^{-4}$) as a function of the equivalent particle diameter, $D$ (m).

$K_{\mathrm{dp}}$ contains the dependence of PRO observable on a series of microphysics parameters, such as the size, the shape and the amount of hydrometeors. The scattering amplitude matrix (**S**, 2×2) provides information of the scattered field after propagating through an ensemble of hydrometeors. It depends on the shape and type of the hydrometeors (e.g. raindrops, pristine ice
crystals, aggregate of ice particles, etc.). The dependence on the shape is often expressed in terms of the axis ratio, i.e. the ratio between the two main axis of the particle. For a perfectly spherical particle, $S_{hh} - S_{vv} = 0$. It also depends on the size of the hydrometeors (generally directly linked to the shape and type). This dependence on the size is proportional to its volume, and therefore it is proportional to the third moment of the $N(D)$. While polarimetric RO do not provide any information on dynamics, the expanding data record provided by ROHP, to directly sense convective ice cloud properties jointly with the
thermodynamic profile, provides a potentially unique addition to the POR.



The water content (WC) is also proportional to the third moment of $N(D)$ as (e.g. Bringi and Chandrasekar, 2001):

$$\mathrm{WC} = 10^{-3}\frac{\pi}{6}\rho \int\limits_{D} D^3 N(D)\mathrm{d}D \tag{3}$$

where $\rho$ is the density in g cm$^{-3}$ and WC is expressed in g m$^{-3}$. Therefore, it is expected that there exists a relationship between $K_{\mathrm{dp}}$ and WC. In fact, the use of $K_{\mathrm{dp}}$ to retrieve WC has been attempted before, for rain (e.g. Jameson, 1985) and for ice (e.g. Vivekanandan et al., 1994; Ryzhkov et al., 1998; Bukovčić et al., 2018). However, in the case of ice and snow, there are additional factors that must be taken into account, namely, the percentage of horizontally oriented particles with respect to those that are randomly oriented, and the associated composition (i.e., dielectric constant of the ice/air media). In the case that all particles were randomly oriented, the $K_{\mathrm{dp}}$ would cancel out regardless of the amount of WC.

In Padullés et al. (2022), the authors showed that the presence of horizontally oriented frozen particles is required in order to explain some of the PAZ PRO observations. In this new paper, we carry out an extensive analysis of the global relationships between water content and $\Delta\Phi$. To do so, a synthetic radio occultation mission is simulated based on ice water properties provided by the Cloudsat data products between 2006-2016. That is, a set of actual radio occultation rays are artificially placed overlaying the Cloudsat curtain-like observations, continuously thorough the whole period. In this way the Cloudsat observations, like radar reflectivity, and its retrievals, like ice water content (IWC), can be analyzed as if they were observed in RO geometry. The statistics gathered from these observations would be equivalent to any other RO mission, if this was able to measure such quantities.

This strategy arises from the fact that there are no coincident observations between the Cloudsat and PAZ, due to different orbit parameters. However, the artificially generated products enable to compare, in a statistical or climatological way, the PAZ $\Delta\Phi$ observations with measurements and retrievals from Cloudsat, such as the IWC.

In this paper we focus on the spatial correlations between the IWC as would have been sensed by RO geometries in Cloudsat measured backgrounds and the $\Delta\Phi$ from PAZ. If these two quantities are related, the spatial patterns should agree and the correlation should be high. This paper is therefore structured as follows: in Section 2 the artificial products based on Cloudsat retrievals and actual RO geometries are described. In Section 3 the correlations between the actual PAZ observations and the generated products are investigated. Finally, in Section 4, forward scattering simulations are used to validate and contextualize the relationships found in the previous section. The paper ends with a discussion on the findings.

## 2 Cloudsat retrievals projected to RO geometries

The Cloudsat satellite was launched in April 28, 2006, and has been operating until August, 2020. It orbits at an approximate height of 715 km in a polar orbit (inclination of 98.2 deg) with an equatorial crossing time of about 1:30 PM. It carries a Cloud Profiling Radar (CPR) operating at 94 GHz (W-band) aimed at observing and characterizing clouds (Stephens et al., 2008). Its high frequency radar is particularly good at sensing the frozen part of clouds and precipitation, whereas it has problems in penetrating to the lower altitudes in the deepest convective regions of precipitating systems (e.g. Battaglia et al., 2007).



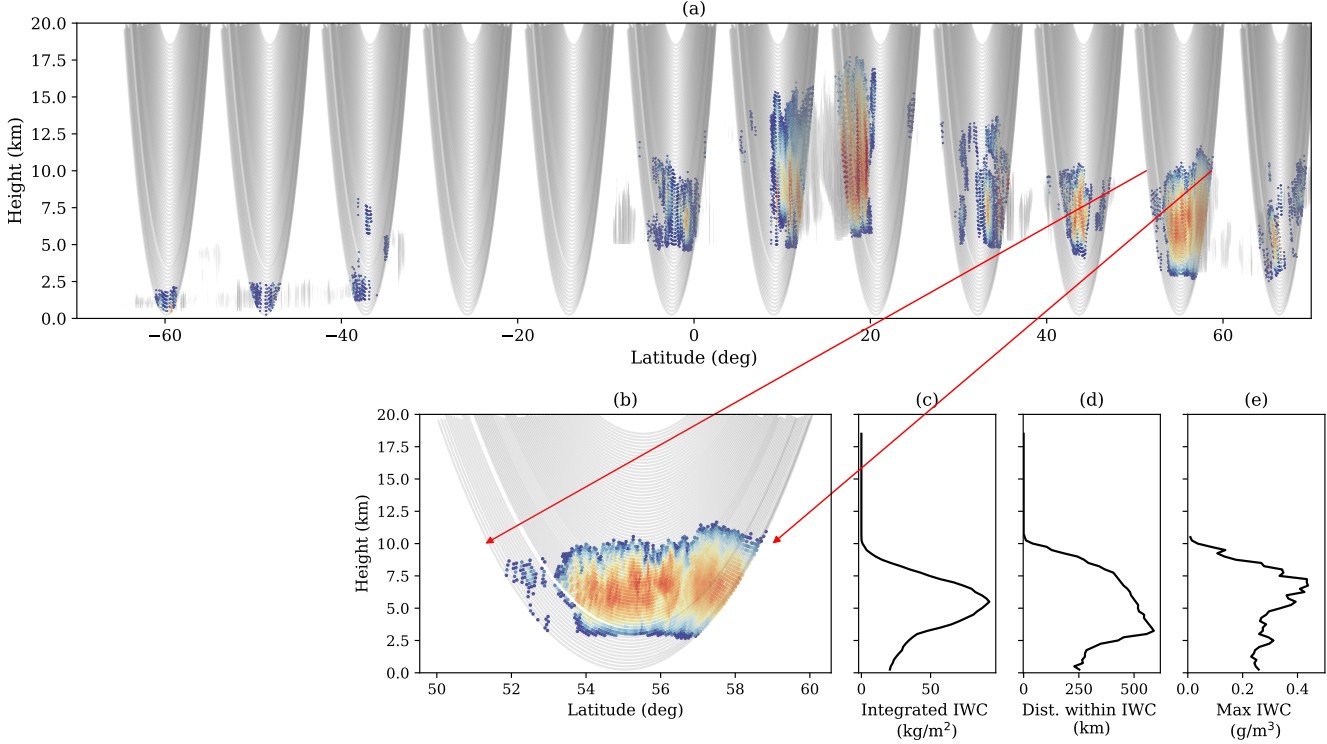

**Figure 1.** Example of half Cloudsat orbit observations (in this case, derived ice water content) with the RO rays overlapped. The Cloudsat orbit granule is 01075. In panel (a), only the Cloudsat ice water content being overlapped by RO rays are colored. IWC not being overlapped by RO rays are in gray-scale. Panel (b) shows a more detailed image of the Cloudsat derived ice water content interpolated into the RO rays, corresponding to one of the slices shown in panel (a) - indicated by the red arrows. In panels (c), (d), and (e) there is shown the corresponding integrated ice water content, distance traveled by the ray within the influence of IWC, and the maximum IWC encountered along the ray, respectively, as a function of each ray tangent height.

The parameters observed and derived from Cloudsat, of interest for this study, consist of the radar reflectivity ($Z_e$) in the 2B-GEOPROF product (Marchand et al., 2008), the IWC and particle size distribution parameters ($N_t$ and $\sigma$) in the 2B-CWC-RO product (Austin et al., 2009). Auxiliary thermodynamic parameters such as temperature and humidity have been obtained from the European Centre for Medium-range Weather Forecasts (ECMWF) model in the ECMWF-AUX products. All these

parameters are obtained with a vertical resolution of 240 m from the surface up to 25-30 km. Since the aim of this study is the comparison with the PRO observations from PAZ, these parameters are transformed to the "RO observing frame", as is defined below.

In brief, radio occultation observations consist of a Low Earth Orbiter (LEO) tracking a Global Navigation Satellite System

(GNSS) emitted signal while it is occulting behind the Earth horizon (e.g. Kursinski et al., 1997; Hajj et al., 2002). This results in the electromagnetic waves from GNSS crossing the different layers of the atmosphere in a tangential way (with respect to





Earth's surface), what is commonly known as a limb sounding observation. One implication of this kind of measurement is that rays cross a long portion of the atmosphere, becoming longer as rays approach the surface. In fact, when rays penetrate below 20 km (region where clouds are expected), their traveled distance can be as large as $\sim 800$ km. In the particular case

of PRO, this has been discussed in Turk et al. (2021). Therefore, direct comparisons between the nadir-looking radar obtained parameters with measurements from RO does not provide relevant information in terms of understanding PRO observations.

The approach followed in this study has been to re-map the Cloudsat observed and derived parameters into the RO geometry. This is accomplished by using a set of actual RO rays and move and rotate them in order to place them overlaid to the curtain-like Cloudsat observations. RO rays often suffer from a drift as the occultation advances (e.g. the collection of rays do not form

a perfectly vertical plane) due to the relative movement of the GNSS satellite and the LEO. However, for this study, the RO rays are collapsed into the Cloudsat vertical plane. Each Cloudsat orbit is split into 1200 km segments (resulting in approximately 29 segments per orbit), and the RO rays are placed in the middle of each of these segments. After that, Cloudsat observed and derived parameters are interpolated into the RO rays. See Fig. 1-(a and b) for an example of the procedure.

This approach permits the consideration of two relevant things. The first one is that one can see the actual extent, and

therefore the influence area, of the cloud structures as sensed by PRO. The second one is that the actual amount of water content along RO rays can be quantified (e.g. see panel c in Fig. 1). Therefore, the integrated ice water content along these RO rays can be compared with the PAZ PRO $\Delta\Phi$ observations. Each RO ray is identified by the height of its lowest point along the ray (i.e. height of what is called the "tangent point"), so the integrated quantities are expressed as function of such height.

Such an exercise is repeated for all Cloudsat orbits between 2006 and 2016, resulting in 684,036 artificial RO observations.

For each of these artificial ROs, the following parameters are stored:

- Location and UTC time of each artificial RO observation. The location is determined by the latitude and longitude of the tangent point that has an altitude of 5 km. The UTC time is the one corresponding to the Cloudsat observation.

- Minimum infra-red ($11\mu$m) brightness temperature ($Tb_{11}$) around a 1 deg radius from the location of the artificial RO, from the National Centers for Environmental Prediction Climate Prediction Center (NCEP CPC) geostationary

satellites observations (Janowiak et al., 2017). This is stored because this quantity is routinely obtained for each PAZ PRO observation (Padullés et al., 2020), and it therefore provides a link between the two datasets.

- The integrated IWC in kg m$^{-2}$ (as a function of tangent height), and the maximum IWC in g m$^{-3}$ encountered along each ray (e.g. Fig. 1-(c and e)).

- The distance each ray traveled within the influence of non-zero water content, in km (e.g. Fig. 1-(d)).

- Additionally, vertical profiles of some thermodynamic parameters at the location of each artificial RO, like the temperature, pressure and specific humidity, are obtained from the ECMWF auxiliary files.

The resulting data are called hereafter the Cloudsat-RO database. In addition to the aforementioned parameters, for a smaller portion of the Cloudsat mission corresponding to 2007, the whole RO planes with the corresponding interpolated parameters are stored. These contain the Cloudsat observed $Z_e$, and the derived IWC and particle size distribution parameters $N_t$ and $\sigma$.



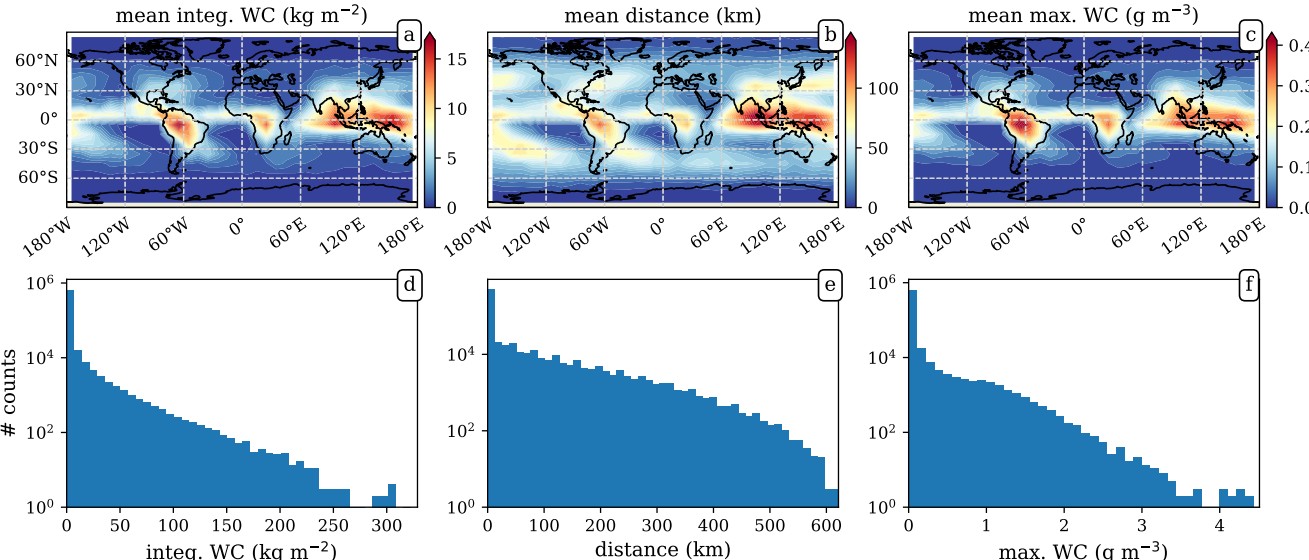

**Figure 2.** Statistics of the Cloudsat-RO database corresponding to the rays whose tangent height is 9 km. Panels (a), (b) and (c) show the mean value climatology for the along-ray integrated IWC, distance (i.e. distance traveled by each ray within the influence of positive IWC), and maximum IWC encountered along the ray, respectively. The used grid is $10 \times 10$ deg. Panels (d), (e), and (f) correspond to the histograms for the total values for the same integrated IWC, distance, and maximum IWC as in the first row.

## 2.1 Statistics

Given that Cloudsat is orbiting on a polar orbit similar to that of PAZ satellite, and that the used RO rays are obtained from real RO events, the statistics gathered from the artificial collocation exercise should resemble the reality observed by PAZ. Furthermore, the integrated parameters along the RO rays mimic the behavior of $\Delta\Phi$ observations, also obtained by PAZ as function of the tangent height. Therefore, to analyze the statistics of the artificial Cloudsat-ROs provides us with valuable information in order to understand the PAZ observations.

Figure 2 shows statistics of the along-ray integrated IWC, distance (here defined as the distance traveled by each ray within the influence of positive IWC), and maximum IWC encountered along each ray, obtained from the Cloudsat-RO database, corresponding to the rays whose tangent height is 9 km. The first thing to observe is the large values for the along-ray integrated IWC (e.g. Fig 2-d, with values up to 250 kg m$^{-2}$), compared with the typical values of vertical ice water path (e.g. Huang et al., 2015). This is due to the long distances that rays travel through IWC. In this geometry, the distance relates more to the size of the storms being observed, as opposed to the vertical water path which relates stronger to their intensity. However, the broad range of distances (see Fig 2-e), reveals the ambiguity of PRO, between the intensity of the storm (or amount of water content) and the distance within the influence of such water content (see Eq. 1).





Another relevant thing is the spatial/geographical patterns. Higher concentrations of IWC at 9 km occur in the tropics (as
expected), specially over land (south America and central Africa) and around the West Pacific warm pool. Regarding the
distance, large values also appear over the mid- and high- latitude oceans. This results agree well with known patters and
previous characterization of storm features, such as in Liu and Zipser (2015), meaning that the RO geometry does not pose a
problem in obtaining these global properties of heavy precipitation systems. Therefore, this database of IWC in RO geometry
is well suited to both help understand the features observed in the PAZ $\Delta\Phi$ and relate them to realistic precipitating systems.

## 3    Relationship between PAZ $\Delta\Phi$ and Cloudsat water content

As it has been discussed in the introduction, both $K_{\mathrm{dp}}$ and WC depend on the third moment of the $N(D)$. Therefore, a
relationship between the two is expected. Being the PRO observable, $\Delta\Phi$, an integral measurement, the corresponding quantity
to compare with would be the integrated WC along the same ray where $\Delta\Phi$ has been measured. However, the PAZ satellite
is in an 6AM/6PM orbit optimized for its primary sensor, which results in few crossover measurements with the GPM and
CloudSat radars. Therefore, the comparison is performed with the built Cloudsat-RO database in statistical terms.

For the $\Delta\Phi$ observations, the whole PAZ mission is used. That is, 152,319 $\Delta\Phi$ profiles between May 10th 2018 and Novem-
ber 30th 2021. All these profiles have been selected so they have passed the initial quality controls and calibration (e.g. Padullés
et al., 2020). PAZ $\Delta\Phi$ data are available from https://paz.ice.csic.es/.

Figure 3 shows the comparison of the mean along-ray integrated IWC with the mean $\Delta\Phi$, as a function of latitude, corre-
sponding to a tangent height of 8 km. It can be seen how the two quantities agree remarkably well when the whole datasets
are considered (panel a). The agreement is specially good over ocean (panel b). In the tropics, the shape of both the along-ray
integrated IWC and $\Delta\Phi$ capture very well the global precipitation signature of the Inter-Tropical Convergence Zone (ITCZ)
(e.g. Marshall et al., 2014; Schneider et al., 2014). Over land (panel c), some features are still recognizable, but the general
agreement is not as good as over ocean.

On a map, the gridded mean of $\Delta\Phi$ can be seen in Figure 4. Panel (a) corresponds to the mean $\Delta\Phi$ at a tangent height of 7 km,
and panel (b) at a tangent height of 9 km. The latter can be compared with Fig. 2-a to investigate whether the spatial patterns
are equivalent or not. In order to quantify the spatial relationship between the two (e.g. PAZ $\Delta\Phi$ and Cloudsat-RO WC), the
mean climatologies are computed on the same grid. The chosen grid is $12 \times 12$ deg, so that spatial patterns of precipitation
arise and there are enough PAZ observations to achieve significant statistics. The values for the mean climatology of integrated
Cloudsat-RO WC and PAZ $\Delta\Phi$ obtained at the same grid cells are compared against each other (Figure 4-c). For the case of
data corresponding to a tangent height of 9 km, the relationship follows a linear trend. The Pearsons correlation coefficient is
0.92 (e.g, $r^2 = 0.86$) therefore exhibiting a robust relationship.

The correlation coefficient for the spatial relationship between the mean integrated Cloudsat-RO WC and mean PAZ $\Delta\Phi$ is
computed for heights between 2 and 17 km. In addition to the mean, the climatologies for the 80th and 90th percentiles are also
computed in order to check if the relationships stands at the higher ends of the distributions. The results are shown in Figure 5
(solid lines). Along with the correlation coefficient, the ratio between the the mean integrated Cloudsat-RO WC and mean PAZ

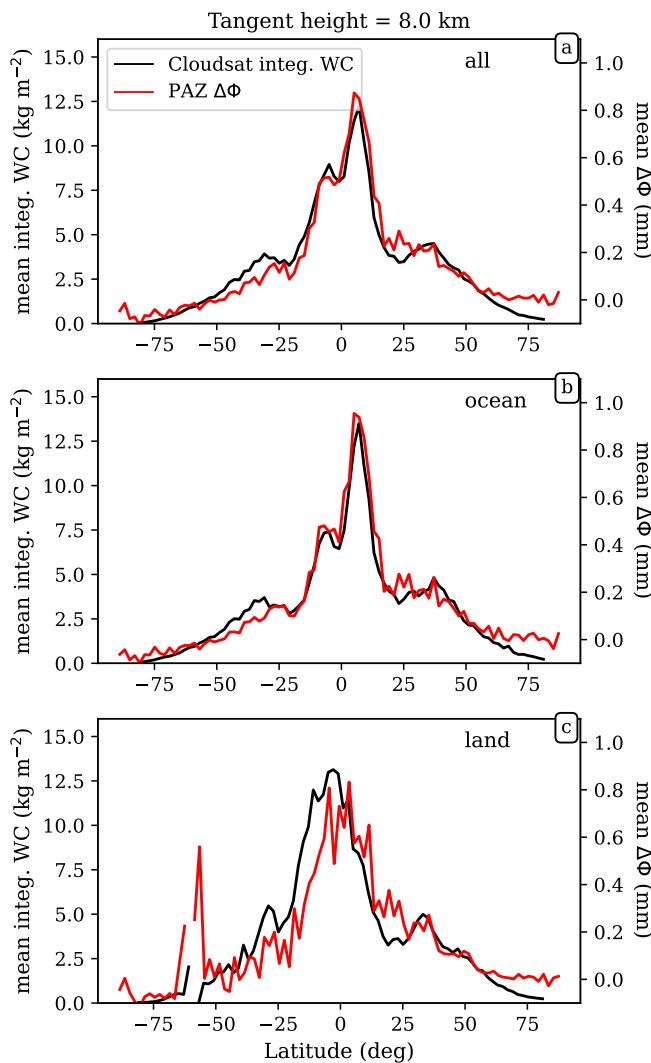

**Figure 3.** Mean values of integrated IWC (left axes) from the Cloudsat-RO dataset, and mean values of observed $\Delta\Phi$ from PAZ (right axes), as a function of latitude, for all data (a), ocean (b) and land (c). Data correspond to tangent heights of 8 km. The mean values are computed for every 2 deg latitude bands.

$\Delta\Phi$ is also computed for all heights (dashed lines in Figure 5). The values for the two first rows of Figure 5 are summarized in Tables 1 and 2.

### 3.1  Correlations

The correlation coefficients in Figure 5 quantify the agreement between the spatial or geographical patterns of the two datasets. High correlations can be understood as the two datasets observing the same kind of precipitation structures. One thing to note

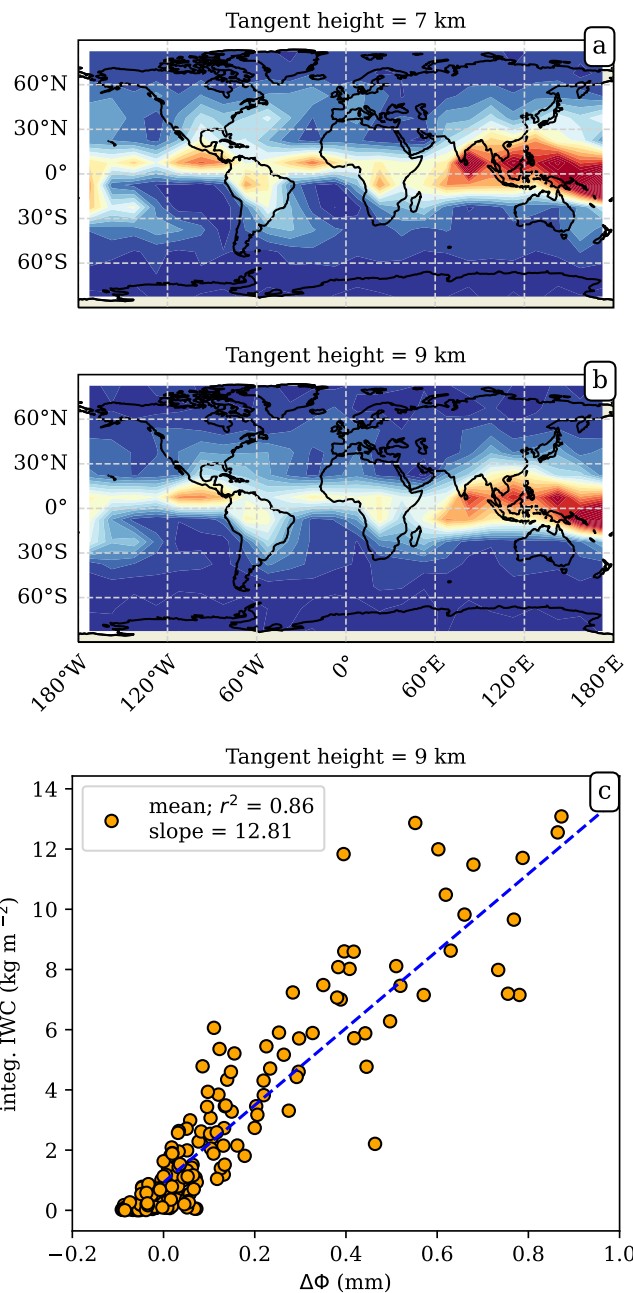

**Figure 4.** Global climatology maps for the mean PAZ $\Delta\Phi$ observations corresponding to a tangent height of 7 km (panel a) and 9 km (panel b). The grid size where the means are computed is 12×12 deg. Panel (c) shows the scatter plot of the mean climatologies of Cloudsat-RO integrated WC vs the PAZ $\Delta\Phi$ obtained at the same grid cell, for the data corresponding to a tangent height of 9 km. Therefore, it is the comparison between panel (b) in this figure and Figure 2-a.





before digging deeper into the results is that the correlation coefficient is computed assuming a linear relationship between the two quantities.

The results, in general (e.g. Figure 5,a-c, and Table 1 - all data considered), show a very high correlation coefficient (i.e. >0.8) between 6 and 12 km. This is particularly high over the oceans, although over land the agreement is also good. This is true for the three statistics being considered, the mean climatology, and the climatologies for the 80th and 90th percentile of the distributions. However, the correlation coefficient corresponding to the mean tends to be slightly higher than the other two.

When the data are split in different regions and surfaces (e.g. Figure 5,d-m), more detailed features can be explored. The first relevant thing is to observe the correlation coefficient in the tropics (e.g. Figure 5,d-f and Table 2), which is high for a wider range of heights, from 2 to 12-13 km. The correlation is specially high over ocean, with values exceeding 0.9 between 5 and 9 km. On the other hand, correlation coefficients over land are not as high as over ocean, although these exceed 0.7 from 2 to 10 km.

It is interesting to note the high correlation at heights expected to be below the freezing height (e.g. 2-5 km in the tropics). Due to the used geometry and retrievals (e.g. IWC from Cloudsat 2B-CWC-RO), it is expected that the ray-points below the freezing level are not contributed with ice water content (because it is liquid), meaning that the simulated integrated IWC from Cloudsat is basically contributed by the portion of the rays above freezing level. Then, the fact that the spatial correlation at the heights below the freezing level is high, indicates that the $\Delta\Phi$ measurements at these regions are mostly contributed by the water content present in the frozen part of the cloud structures.

On the contrary, this behavior is not observed outside the tropics. For the regions corresponding to 70S-30S and 30N-70N (e.g. Figure 5,g-l) the correlation coefficient is very low below 5 km. It is worth noting that in the southern hemisphere latitude range between 30-70S, cases over land are almost nonexistent. This explains the large dispersion in Figure 2-i panel. The low correlation coefficients in the southern oceans are further discussed in Sect. 5.

## 3.2 Ratios

For the regions and heights where the correlation coefficient is high (bold numbers in Tables 1 and 2), the ratio between the Cloudsat-RO IWC and PAZ $\Delta\Phi$ (i.e. ratio = IWC/$\Delta\Phi$) provides empirical information on the relationship between the two quantities. In general, the ratio is low (e.g. ratio < 10) in the low layers (below 5 km), increases between 5 and 12 km, and decreases again for higher layers.

Unlike the correlation, the ratio between the IWC and $\Delta\Phi$ is sensitive to all contribution to IWC. That is, in the lower layers where the contribution from liquid water content is missing in the Cloudsat-RO dataset, the ratio does not fairly represent the relationship between WC and $\Delta\Phi$. This effect could also appear in the uppermost layers, where Cloudsat high sensitivity might be accounting for more IWC than that contributing to $\Delta\Phi$. In addition, the ratio is highly sensitive to the % of horizontally oriented with respect to the amount of random oriented frozen particles. This is further developed in Sect. 4.

Looking at the actual numbers and focusing only in those layers where the correlation coefficient is high (e.g. cc > 0.8), the ratio between the IWC and $\Delta\Phi$ ranges between 5 - 20 for the mean climatological values in the tropics, whereas the ratios

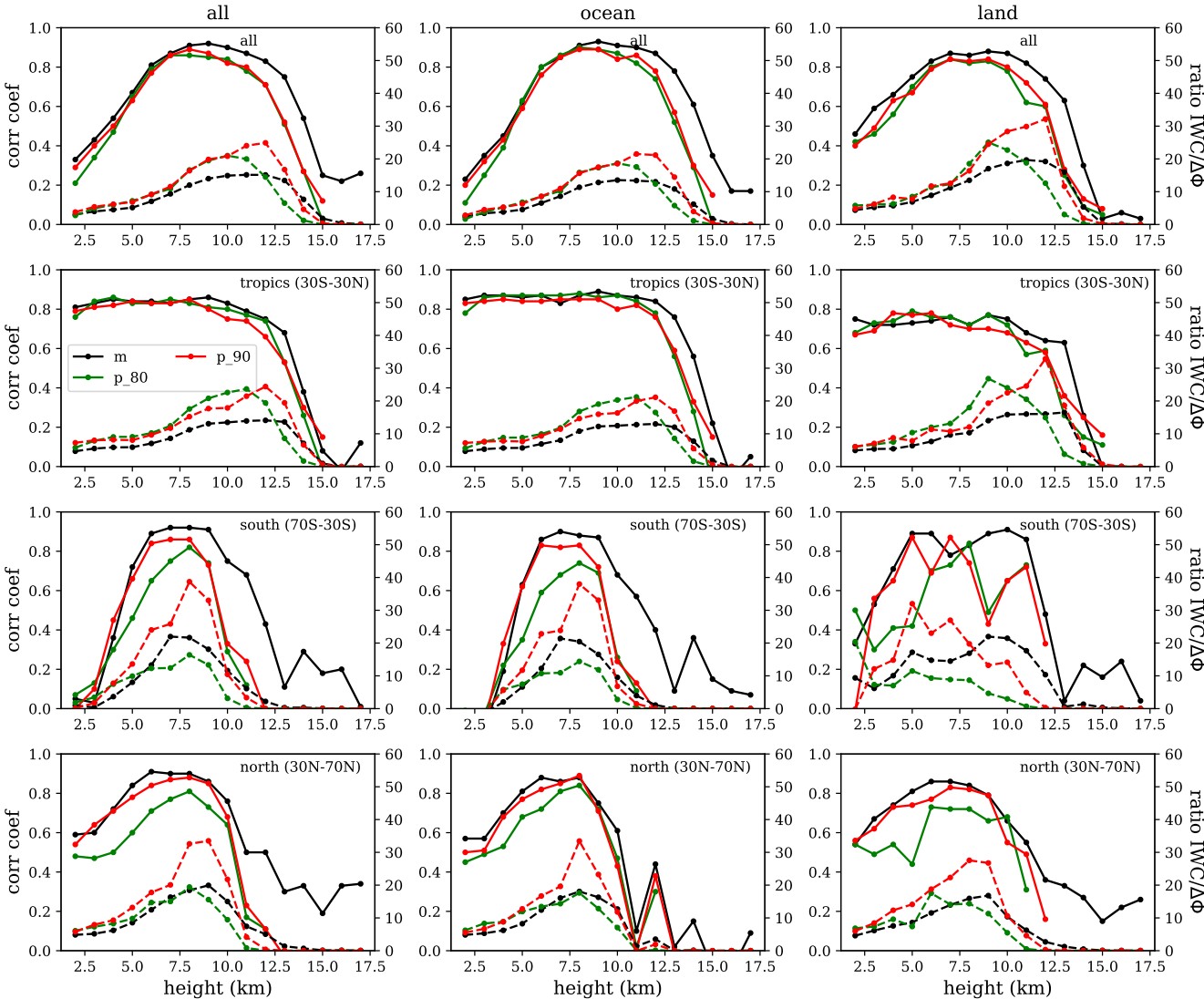

**Figure 5.** Correlation coefficients (solid lines - left axes) and ratio (dashed lines - right axes) between Cloudsat-RO IWC and PAZ $\Delta\Phi$, as a function of height (x-axis), for different areas across the globe (all globe, tropics, southern extra-tropics, and northern extra-tropics), and different surfaces (all, ocean, land). The black lines correspond to the mean climatologies, the green lines to the 80th percentile and the red lines to the 90th percentile climatology.

reach values of around 25 outside tropics. It can also be seen that the ratio generally increase with height. At the higher ends of the distributions (e.g. 80th and 90th percentiles), the ratios are larger than for the means, but follow similar patterns.





**Table 1.** Values for the correlation coefficient and ratio between IWC and $\Delta\Phi$ (top and bottom value in each cell, respectively), for different statistics (first column), different regions (second column) and different heights (first row). Data corresponds to the whole globe. Values in bold highlight correlation coefficients higher than 0.8.

| | height (km) | 3 | 4 | 5 | 6 | 7 | 8 | 9 | 10 | 11 | 12 | 13 | 14 | 15 |
|---|---|---|---|---|---|---|---|---|---|---|---|---|---|---|
| mean | all | 0.43 | 0.54 | 0.67 | **0.81** | **0.87** | **0.91** | **0.92** | **0.9** | **0.87** | **0.83** | 0.75 | 0.54 | 0.25 |
| | | 3.98 | 4.52 | 5.15 | 7.05 | 9.35 | 12.01 | 13.99 | 14.9 | 15.15 | 15.12 | 13.41 | 7.67 | 1.86 |
| | ocean | 0.35 | 0.45 | 0.62 | 0.8 | **0.85** | **0.91** | **0.93** | **0.91** | **0.9** | **0.87** | 0.78 | 0.61 | 0.35 |
| | | 3.42 | 3.81 | 4.55 | 6.54 | 8.63 | 11.36 | 12.81 | 13.54 | 13.38 | 13.05 | 10.84 | 6.2 | 1.77 |
| | land | 0.59 | 0.66 | 0.75 | **0.83** | **0.87** | **0.86** | **0.88** | **0.87** | **0.82** | 0.74 | 0.63 | 0.3 | 0.03 |
| | | 5.21 | 5.72 | 6.88 | 8.89 | 11.19 | 13.48 | 17.04 | 18.63 | 19.68 | 19.21 | 15.98 | 5.35 | 0.25 |
| p80 | all | 0.34 | 0.47 | 0.65 | 0.79 | **0.86** | **0.86** | **0.85** | **0.84** | 0.78 | 0.71 | 0.51 | 0.27 | 0.02 |
| | | 4.93 | 5.99 | 7.21 | 9.12 | 10.92 | 16.62 | 19.48 | 20.99 | 19.98 | 14.47 | 6.51 | 1.24 | 0.01 |
| | ocean | 0.25 | 0.39 | 0.63 | 0.8 | **0.86** | **0.9** | **0.89** | **0.87** | **0.82** | 0.74 | 0.52 | 0.29 | 0.01 |
| | | 3.8 | 5.22 | 6.78 | 8.52 | 10.2 | 15.86 | 17.33 | 18.69 | 17.58 | 12.35 | 5.77 | 1.14 | 0.0 |
| | land | 0.46 | 0.56 | 0.7 | 0.8 | **0.84** | **0.82** | **0.83** | 0.78 | 0.62 | 0.6 | 0.25 | 0.09 | 0.05 |
| | | 6.06 | 6.29 | 8.5 | 11.36 | 12.32 | 18.54 | 25.06 | 22.7 | 18.78 | 12.6 | 3.03 | 0.31 | 0.01 |
| p90 | all | 0.4 | 0.5 | 0.63 | 0.77 | **0.86** | **0.89** | **0.87** | **0.82** | 0.8 | 0.71 | 0.52 | 0.27 | 0.12 |
| | | 5.39 | 6.18 | 6.84 | 9.19 | 11.55 | 16.39 | 19.84 | 20.83 | 24.03 | 24.9 | 16.75 | 4.64 | 0.36 |
| | ocean | 0.32 | 0.43 | 0.59 | 0.76 | **0.85** | **0.89** | **0.89** | **0.84** | **0.86** | 0.78 | 0.57 | 0.3 | 0.15 |
| | | 4.51 | 5.26 | 6.29 | 8.59 | 10.98 | 15.63 | 17.49 | 18.52 | 21.51 | 21.17 | 14.46 | 3.98 | 0.4 |
| | land | 0.49 | 0.63 | 0.67 | 0.79 | **0.84** | **0.83** | **0.84** | 0.8 | 0.72 | 0.61 | 0.28 | 0.13 | 0.08 |
| | | 6.21 | 8.3 | 7.78 | 11.72 | 12.54 | 16.36 | 24.48 | 28.37 | 29.88 | 32.17 | 11.64 | 1.91 | 0.2 |

## 3.3 Heavy precipitating cases and differences between ocean and land

### 3.3.1 Results for a subset of heavy precipitating cases

Tu further examine the relationship between Cloudsat-RO WC and PAZ $\Delta\Phi$, in this section we focus on a subset of tropical (within $\pm30°$ latitude) heavy precipitating cases. The cases (both the artificial RO and actual PAZ observations) where the $Tb_{11} < 205\,\mathrm{K}$ are selected. For the artificial cases, the 2-dimensional histogram (where y-axis is the height, and x-axis is the value being represented) of Cloudsat radar reflectivity, distance within the influence of non-zero water content, the IWC, and the two parameters $N_t$ and $\sigma$, are shown in Figure 6. These quantities correspond to those that have been integrated into the





**Table 2.** Same as in Table 1, but for the tropics (30S-30N).

| | height (km) | 3 | 4 | 5 | 6 | 7 | 8 | 9 | 10 | 11 | 12 | 13 | 14 | 15 |
|---|---|---|---|---|---|---|---|---|---|---|---|---|---|---|
| mean | all | **0.83** | **0.85** | **0.84** | **0.84** | **0.83** | **0.85** | **0.86** | **0.83** | 0.79 | 0.75 | 0.68 | 0.38 | 0.08 |
| | | 5.5 | 5.85 | 5.9 | 7.05 | 8.6 | 11.21 | 13.07 | 13.48 | 13.91 | 14.1 | 13.62 | 7.07 | 0.94 |
| | ocean | **0.87** | **0.87** | **0.86** | **0.87** | **0.83** | **0.87** | **0.89** | **0.87** | **0.86** | **0.84** | 0.76 | 0.56 | 0.22 |
| | | 5.31 | 5.66 | 5.69 | 6.89 | 8.02 | 10.8 | 12.23 | 12.47 | 12.78 | 13.0 | 12.02 | 7.75 | 1.82 |
| | land | 0.72 | 0.72 | 0.73 | 0.74 | 0.76 | 0.72 | 0.77 | 0.75 | 0.68 | 0.64 | 0.63 | 0.26 | 0.0 |
| | | 5.36 | 5.43 | 6.36 | 7.64 | 9.63 | 10.32 | 14.0 | 15.85 | 16.01 | 16.03 | 16.5 | 4.94 | 0.05 |
| $p_{80}$ | all | **0.84** | **0.86** | **0.83** | **0.83** | **0.85** | **0.83** | **0.81** | 0.8 | 0.77 | 0.74 | 0.53 | 0.26 | -0.01 |
| | | 7.84 | 9.04 | 9.08 | 10.22 | 12.42 | 17.57 | 20.81 | 22.58 | 23.67 | 19.41 | 8.59 | 1.68 | -0.01 |
| | ocean | **0.86** | **0.87** | **0.87** | **0.87** | **0.87** | **0.88** | **0.86** | **0.87** | **0.84** | 0.78 | 0.56 | 0.28 | -0.08 |
| | | 7.56 | 8.83 | 8.83 | 9.92 | 11.79 | 16.81 | 19.05 | 20.31 | 21.23 | 16.5 | 8.59 | 1.61 | -0.03 |
| | land | 0.73 | 0.74 | 0.79 | 0.76 | 0.76 | 0.72 | 0.77 | 0.72 | 0.57 | 0.59 | 0.26 | 0.15 | 0.11 |
| | | 6.72 | 7.48 | 10.36 | 11.98 | 13.14 | 17.99 | 26.81 | 24.01 | 20.52 | 14.93 | 3.76 | 0.88 | 0.06 |
| $p_{90}$ | all | **0.81** | **0.82** | **0.84** | **0.83** | **0.83** | **0.85** | 0.8 | 0.75 | 0.74 | 0.66 | 0.53 | 0.3 | 0.15 |
| | | 7.99 | 8.17 | 8.06 | 9.66 | 11.7 | 15.23 | 17.69 | 17.88 | 21.46 | 24.39 | 19.44 | 6.63 | 0.62 |
| | ocean | **0.84** | **0.85** | **0.84** | **0.84** | **0.85** | **0.85** | **0.85** | 0.8 | **0.82** | 0.76 | 0.59 | 0.33 | 0.15 |
| | | 7.59 | 7.8 | 7.64 | 9.35 | 11.41 | 14.64 | 15.97 | 16.34 | 19.97 | 21.12 | 16.89 | 5.48 | 0.57 |
| | land | 0.69 | 0.78 | 0.77 | 0.78 | 0.72 | 0.7 | 0.7 | 0.68 | 0.63 | 0.58 | 0.36 | 0.25 | 0.16 |
| | | 7.05 | 8.78 | 7.84 | 11.32 | 10.72 | 12.07 | 19.33 | 22.49 | 24.57 | 32.85 | 18.64 | 5.83 | 0.7 |

realistic RO plane (see Sect 2 for details). In addition, the mean and the 85th percentile of the integrated WC as a function of
the tangent height for the selected cases are also shown. These results are further split in over ocean and over land cases (top
and bottom rows). For the selected actual PAZ observations, the mean and the 85th percentile of the $\Delta\Phi$ as a function of the
tangent height are also shown, equally split in over ocean and over land. For representation purposes, the $\Delta\Phi$ mean and 85th
percentile are multiplied by a factor of 15.

In the results shown in Figure 6 two important things can be observed. First, that there is a clear difference between the
$\Delta\Phi$ observations over ocean and over land. While the mean integrated water content (and the corresponding 85th percentile)
are relatively similar over ocean and over land (black and gray lines), there is a more obvious difference between the $\Delta\Phi$
observations over ocean and land. The fact that IWC is similar should lead to similar $\Delta\Phi$, but $\Delta\Phi$ observed by PAZ tends to
be diminished over land.





The second thing is that the $\Delta\Phi$ and WC profiles over ocean agree well within each other and with the shape of the 2D
histograms of radar reflectivity, distance within non-zero water content, and retrieved water content. The agreement is specially
good above 6 km. Below that height, $\Delta\Phi$ is probably being importantly contributed by liquid water content and therefore $\Delta\Phi$
profile remains relatively higher than the cloudsat retrieved along-ray integrated IWC, which suffers an important drop in the
lower layers.

The results over ocean confirms the robust relationship between the $\Delta\Phi$ and the along-ray integrated IWC, even for the most
extreme cases. The differences between ocean and land deserve a further discussion.

### 3.3.2 Differences between ocean and land

Cloudsat measured and derived products shown in Figure 6 do not seem to be as different as to induce the large differences
observed between the over ocean and over land $\Delta\Phi$ profiles. This is further examined by plotting the cumulative distribution
functions of these five same products for two fixed heights. This is shown in Figure 7. Some differences are visible (e.g. larger
reflectivity over land than over ocean, larger WC also over land) specially at 12 km. However, at 8 km these differences do not
appear to be as significant, but the $\Delta\Phi$ differences are.

These differences also reinforce the hints provided by the lower correlation coefficients (Sect. 3.1) that the relationship
between $\Delta\Phi$ and IWC over land is more complex than over ocean. The hypotheses for this complexity are developed in the
discussion in Sect. 5

### 250  4   Forward scattering simulations of ice and snow

The correlation coefficients determined in Sect. 3.1 indicate a robust relationship between $\Delta\Phi$ and IWC, specially over tropical
ocean. However, the meaning of the ratios between IWC and $\Delta\Phi$ depend on more factors. The aim of this section is to
determine whether the ratios in Sect. 3.2 are physically meaningful or not. That is, to determine if the ratios are compatible
with the characteristics of hydrometeors that are known to be present in clouds.

For this, the idea is to compute the $K_{dp}$ - IWC relationship for a series of different hydrometeors and ice particle habits.
This relationship can then be compared with the ratios in Fig. 5 and in Tables 1 and  2. To perform such comparison, first,
the scattering amplitude matrix **S** is computed for a set of different single hydrometeors and ice particle habits. Then, a set of
particle size distributions from one week of Cloudsat observations are used to generate the corresponding $K_{dp}$ and IWC using
Eqs. 2 and 3. Note that for this study, only single particle scattering is considered.

### 260  4.1   Hydrometeors and ice particle habits

The forward scattering simulations used for this study have been done using Rayleigh approximation. This formulation as-
sumes that the particles can be approximated as oblate spheroids, with a certain axis ratio and effective density. The list of
hydrometeors and ice particle habits that have been used are pristine ice crystals, aggregates of pristine ice, and wet snow. The
adequacy of using Rayleigh approximations is justified by the long wavelength of GNSS signals, i.e. $\sim$190.3 mm, much larger



**Figure 6.** Two dimensional histograms (being y-axis height, and x-axis the represented quantity) for the Cloudsat radar reflectivity ($Z_e$) (a, f), the distance within the influence of non-zero water content (b, g), the Cloudsat retrieved IWC (c, h), the $N_t$ parameter of the particle size distribution (PSD) (d, i) and the sigma parameter of the PSD (e, j) for the subset of cases where the $Tb_{11} < 205\,\mathrm{K}$. The represented quantities correspond to those interpolated into the RO plane. Further details provided in Sect 2. The top row correspond to cases over ocean, and the bottom row to cases over land. Yellow (red-dashed) lines correspond to the mean (85th percentile) $\Delta\Phi$ as a function of tangent height from the actual PAZ measurements - here following the top-axis and multiplied by a factor of 15. Black (dashed-gray) lines correspond to the mean (85th percentile) of the integrated IWC - following also the top-axis. The $\Delta\Phi$ and WC vertical profiles are repeated in all panels (only differing over ocean vs over land).

than the typical size of frozen hydrometeors. However, for some pristine ice crystals and aggregates, a comparison using the Discrete Dipole Approximation (DDA, Draine and Flatau, 1994) has been performed. For Rayleight approximations, the for-





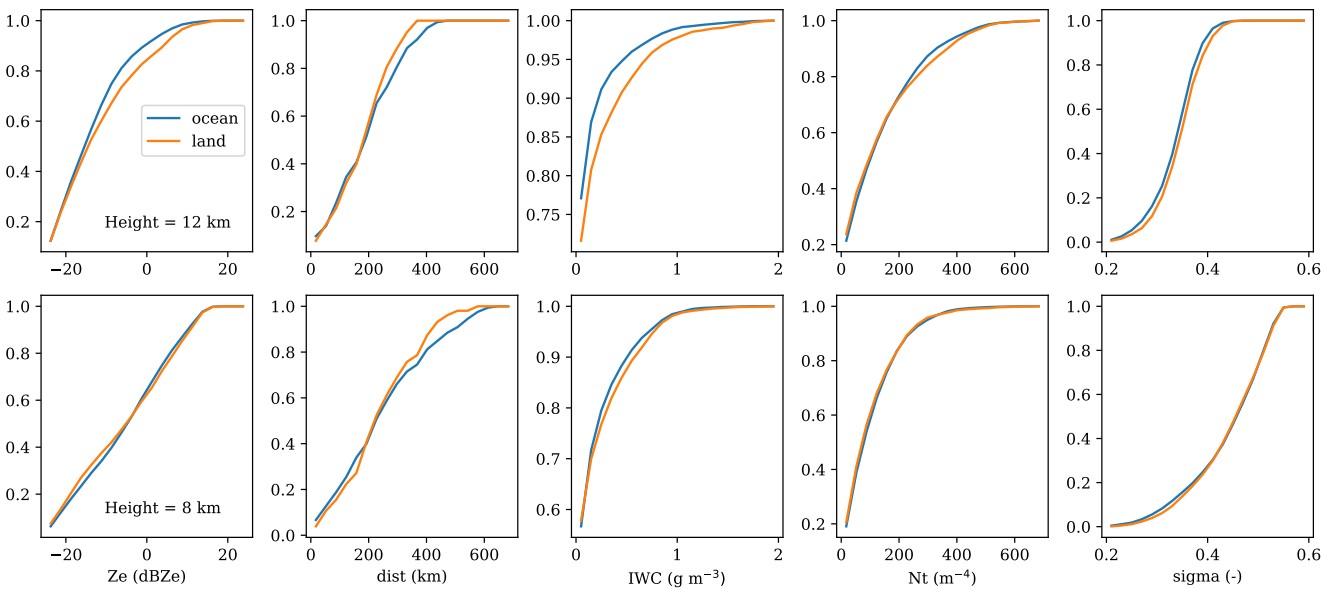

**Figure 7.** Cumulative distribution functions for the quantities shown in Figure 6, for two fixed heights (12 km, first row; 8 km, second row) and for the cases over ocean (blue) and over land (orange).

mulations in Ryzhkov et al. (2011) and in Bringi and Chandrasekar (2001) have been followed. That is, the forward scattering amplitude co-polar components are computed using:

$$S_{\mathrm{hh,vv}} = \frac{\pi^2 D^3}{6\lambda^2} \frac{1}{L_{\mathrm{hh,vv}} + \dfrac{1}{\epsilon - 1}} \tag{4}$$

where $\epsilon$ is the dielectric constant, and $L_{\mathrm{hh,vv}}$ are the shape parameters:

$$L_{\mathrm{hh}} = \frac{1+f^2}{f^2}\left(1 - \frac{\arctan(f)}{f}\right), \quad f = \sqrt{\frac{1}{ar^2} - 1} \tag{5}$$

$$L_{\mathrm{vv}} = \frac{1 + L_{\mathrm{hh}}}{2} \tag{6}$$

where $ar$ is the axis ratio of the particle. For the dielectric constant, the Maxwell-Garnett formula is used in order to account

for the effective density of the particles as mixtures of ice, air, and water (Maxwell Garnett, 1904).

The results of the forward scattering simulations are shown in Fig. 8. The $S_{\mathrm{hh}} - S_{\mathrm{vv}}$ as a function of the equivalent diameter for different types of frozen particles are shown in panel a, while the relationship of $K_{\mathrm{dp}}$ with water content for the same particles is shown in panel b. Below there are the specific details for each of the hydrometeors and habits used in this study.





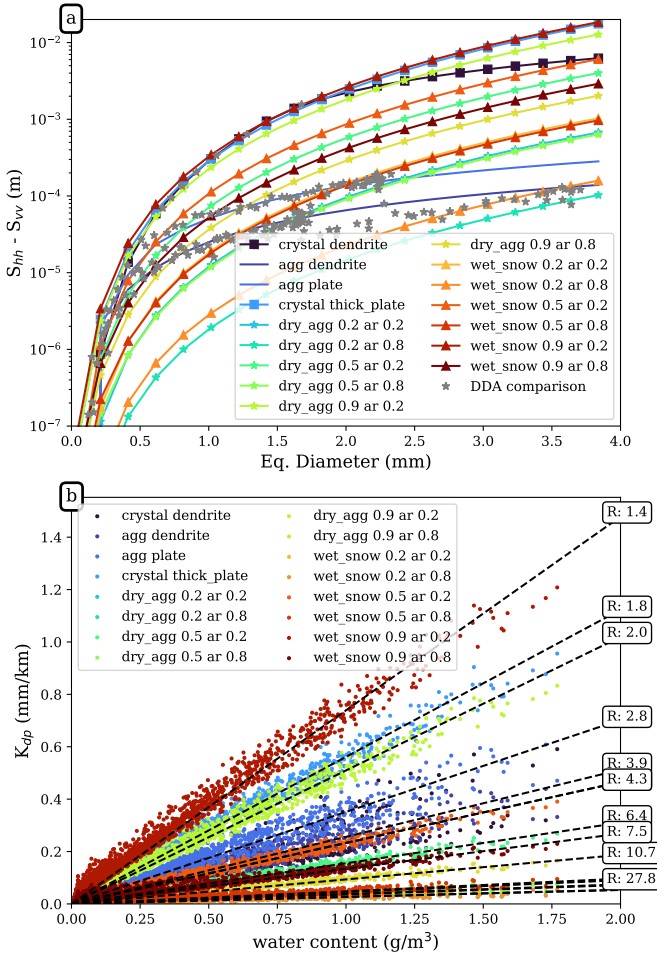

**Figure 8.** Results of the single particle forward scattering computations. (a) Difference between the co-polar components of the scattering amplitude matrix **S** ($S_{\mathrm{hh,vv}}$) as a function of the equivalent melted diameter for a series of hydrometeors and ice particle habits (as indicated in the legend). For dry aggregates and wet snow, the first number indicates the effective density and the second one the axis ratio. See Sect. 4.1 for more details on the particles. (b) $K_{\mathrm{dp}}$ as a function of water content obtained using Eqs. 2 and 3 using common PSD obtained from one week of Cloudsat observations. For each type of particle, the mean ratio between WC and $K_{\mathrm{dp}}$ is shown in the right side of panel (b).

### 4.1.1 Pristine ice crystals

The simplest type of particles used in this study are pristine ice dendrites and thick plates. For these, both the Rayleigh approximation and the exact DDA computations are performed. For the Rayleigh computation, the density of solid ice is used, and the values in Ryzhkov et al. (2011, Table 1) are used for the axis ratio. For the computation using DDA, exact shapes from Liu (2008) are used with the ADDA code (Yurkin and Hoekstra, 2011). For both methods, a fully horizontal orientation of the particles is assumed. The results using the Rayleigh approximation are shown in Fig. 8 (black squares-line for dendrites and





cyan squares-line for the thick plates). The gray stars (ADDA results) on top of the Rayleigh results show the good agreement

between the two methods for these kind of particles at L-band.

### 4.1.2  Aggregates

For more complex shapes, we use aggregates of pristine ice particles. For the Rayleigh approximation, an spheroid of air filled

with portions of ice is used, following Ryzhkov et al. (2011) and taking different values for the effective density and axis ratio.

The results are shown in Fig. 8, for effective densities of 0.2, 0.5 and 0.9, and axis ratios of 0.2 and 0.8 (an axis ratio of 1 would

be a perfect sphere). Note that using an effective density of 0.9 means that the spheroid is filled with ice, approaching a pristine

ice particle. The results using the Rayleigh method for fixed densities and fixed axis ratios for the whole range of equivalent

diameters are called dry aggregates and are shown in Fig. 8 as star-lines.

For comparison purposes, exact computations of aggregates are also performed. To construct the shapes to be used with

ADDA, the approach in Leinonen and Moisseev (2015) for dendrites and plates is followed. The only difference is that in this

paper, the aggregate particles, once generated, are forced to lie in the most horizontally oriented plane as possible (i.e. the

particles are rotated so that the longest dimension of the aggregated particle corresponds to the horizontal axis, and the shortest

to the vertical). Like for the results in Sect. 4.1.1, the results obtained here with DDA are shown in Fig 8-a as gray stars, and

these also fall on top of the solid lines representing the results of the Rayleigh approximation. In the comparison with the exact

shapes, the Rayleigh approximation results are obtained by varying the density and axis ratio with the equivalent diameter,

resembling the realistic shapes used for DDA (solid lines in Fig. 8).

### 4.1.3  Wet snow

Finally, wet snow aims to represent frozen particles at the initial stages of melting. For this, the same spheroids as in the

previous subsection are used, but for this case a mass faction of liquid water of a 10% of the total mass is used, in addition to

pure ice and air, to compute $\epsilon$. The presence of water in the particle enhances $\epsilon$ which in turn increases the $K_{\mathrm{dp}}$ with respect to

the dry aggregates with same parameters (of equivalent density and $ar$). For this case, only results for Rayleigh approximation

are computed, and no exact computations have been performed.

The results are shown in Fig. 8 as triangle-lines for different values of effective density and axis ratio.

### 4.2  Orientation angle distribution

The results shown in Fig. 8-a corresponding to Sect. 4.1 are computed using single particle scattering and forcing the different

particles to be horizontally oriented. For the exact computations using DDA, this is accomplished by placing the particle lying

on the x-y plane on a x,y,z typical Cartesian plane (assuming an incident field in the y-direction), and averaging the results

obtained by rotating the z axis. The corresponding relationships between $K_{\mathrm{dp}}$ and WC shown in Fig. 8-b (square boxes over

the right axis) are therefore for a collection of particles of the same type contributing to the whole water content and fully


horizontally oriented. This is unlikely to be the case in real clouds and storms, and in fact, most of the ratios reported in Fig. 8-b are smaller than most of the observed ones (e.g. Fig. 5) and reported in Tables 1 and 2.

In real scenarios, the orientation of these particles is more complex. To account for different orientation angles, we assume a Gaussian distribution of tilt (or canting) angles centered at $0°$ (hence mean angle $\beta = 0°$) with a certain standard deviation $\sigma$. This implies that varying $\sigma$ we can range from total horizontal orientation ($\sigma = 0°$) to completely random orientation (large

$\sigma$). To keep the computations simple, we can use the horizontal orientation values and multiply them by a factor that accounts for such canting (Oguchi, 1983):

$$K_{dp}^{\sigma} = \frac{1 + e^{-2\sigma^2} \cos(2\beta)}{2} e^{-2\sigma^2} K_{dp}^{\sigma=0} \tag{7}$$

The results for the different $\sigma$ angles applied to the results of Fig. 8-b are shown in Table 3. The particle types have been sorted by the value of the ratio for $\sigma = 0°$. For each particle type, computations using different tilt angles are performed, and

the ratios with a value within the range [5 - 25] are highlighted in bold. These values are the ones agreeing with the observed values in Sect. 3 (e.g. Fig. 5 and Tables 1 and 2). In this results, it can be seen how for more pristine and thin particles, certain canting angle is required (e.g. fully horizontal orientation is overestimating the observed $K_{dp}$), whereas for more complex particles, these are required to stay more horizontally oriented to better match the actual data. It can also be observed that very low density particles with high axis ratios (i.e. approaching empty spheres) alone cannot explain the observations.

**5   Discussion and Conclusions**

The relationship between the PRO observable $\Delta\Phi$ and ice water content has been investigated in a global and statistical (or climatological) way. For this purpose, the measurements and retrievals from the Cloudsat mission have been used. These have been re-mapped into the RO observation geometry so that comparisons take into account the important features of the observation geometry of PRO. An important one is the long distances that rays traveling from the GNSS satellites to the

receivers in Low Earth Orbit spend in the lower layers of the troposphere. Such re-mapping allows an evaluation of the geometry itself. It can be assessed whether a limb-sounding measurement like PRO is able to capture important features of precipitation or not. The results in Fig. 2 show the agreement between the climatology of the along-ray integrated ice water content (and related products such as the distance rays traveled within areas of non-zero ice water content, or the maximum ice water content encountered per ray) with known and previously studied heavy precipitation features (e.g. Liu and Zipser, 2015). This

is further confirmed with the results in Fig. 3 (showing the mean climatology as function of latitude for water content, solid black line), which agrees very well with the signature of the ITCZ. Therefore, the good climatological agreement enables the use of the Cloudsat-based artificially collocated RO database of along-ray integrated ice water content for understanding the PRO observations of precipitating cloud structures. For this study, only ice water content retrievals are used. The use of ice only retrievals is justified because Cloudsat observations in the liquid region of deep cloud structures (such as those specifically

targeted by the ROHP experiment) may be degraded due to the high frequency and penetration issues. However, the missing liquid part may have its implications, specially outside the tropics (as will be discussed below).





**Table 3.** Ratio between IWC and $K_{\mathrm{dp}}$ for the different hydrometeors shown in Fig. 8, when a certain standard deviation is assumed in the orientation distribution (Gaussian distribution centered at $0°$ and $\sigma$) $\sigma = 0°$ represents the scenario where all particles are horizontally oriented, while for large $\sigma$ all particles would be randomly oriented (see Eq. 7). In bold, the ratios within the range [5 - 25], in agreement with the observed ratios in Fig. 5 and Tables 1 and 2.

| | standard deviation of the Gaussian distribution of orientations, $\sigma$ (°) | | | | | | | | |
| --- | --- | --- | --- | --- | --- | --- | --- | --- | --- |
| | 0 | 10 | 20 | 30 | 40 | 50 | 60 | 70 | 80 |
| wet snow 0.9 ar 0.2 | 1.4 | 1.5 | 1.9 | 3.0 | **5.2** | **10.2** | **21.8** | >50 | >50 |
| crystal thick plate | 1.8 | 2.0 | 2.5 | 3.9 | **6.9** | **13.4** | 28.7 | >50 | >50 |
| dry agg 0.9 ar 0.2 | 2.0 | 2.1 | 2.8 | 4.3 | **7.6** | **14.8** | 31.7 | >50 | >50 |
| agg plate | 2.8 | 3.1 | 4.1 | **6.2** | **11.0** | **21.4** | 45.9 | >50 | >50 |
| agg dendrite | 3.9 | 4.3 | **5.6** | **8.6** | **15.0** | 29.4 | >50 | >50 | >50 |
| wet snow 0.5 ar 0.2 | 4.3 | 4.7 | **6.1** | **9.4** | **16.5** | 32.3 | >50 | >50 | >50 |
| crystal dendrite | 4.3 | 4.7 | **6.1** | **9.4** | **16.5** | 32.4 | >50 | >50 | >50 |
| dry agg 0.5 ar 0.2 | **6.4** | **7.1** | **9.2** | **14.1** | **24.8** | 48.5 | >50 | >50 | >50 |
| wet snow 0.9 ar 0.8 | **7.5** | **8.2** | **10.7** | **16.3** | 28.7 | >50 | >50 | >50 | >50 |
| dry agg 0.9 ar 0.8 | **10.7** | **11.7** | **15.3** | **23.4** | 41.1 | >50 | >50 | >50 | >50 |
| wet snow 0.5 ar 0.8 | **20.7** | **22.7** | 29.7 | 45.5 | >50 | >50 | >50 | >50 | >50 |
| wet snow 0.2 ar 0.2 | **22.1** | 24.2 | 31.6 | 48.5 | >50 | >50 | >50 | >50 | >50 |
| dry agg 0.5 ar 0.8 | 26.8 | 29.3 | 38.3 | >50 | >50 | >50 | >50 | >50 | >50 |
| dry agg 0.2 ar 0.2 | 27.8 | 30.5 | 39.8 | >50 | >50 | >50 | >50 | >50 | >50 |
| wet snow 0.2 ar 0.8 | 37.1 | 40.6 | >50 | >50 | >50 | >50 | >50 | >50 | >50 |
| dry agg 0.2 ar 0.8 | 37.6 | 41.2 | >50 | >50 | >50 | >50 | >50 | >50 | >50 |

PRO observable $\Delta\Phi$ is the integrated $K_{\mathrm{dp}}$ along each RO ray. Both $K_{\mathrm{dp}}$ and water content are proportional to the third moment of the $N(D)$, and therefore a relationship between them is to be expected. The relationship is investigated by evaluating the correlation coefficient between the geographical patterns of the high and low concentrations of $\Delta\Phi$ and along-ray inte-

grated IWC, split in different regions and heights. Overall, correlation coefficients (e.g. Fig. 5) are high for the heights where frozen particles are expected (which changes by region, e.g. tropics vs. outside tropics). When the higher ends of the assumed distribution are evaluated (i.e. the 80th and 90th percentiles), similar behavior is observed. When the results are inspected in more detail (detailed in Tables 1 and 2) differences are observed between the tropics and mid-latitudes, and between ocean and land. Tropical oceans is where the correlation coefficients maximize.

The main observed difference between tropics and mid-latitudes is the drop in the correlation coefficient for the lowest heights. The difference in high altitudes is easily explained by the different maximum height of clouds, which is higher in the tropics. One important thing to take into account when focusing in the lower heights is that the ice water content retrieval from the Cloudsat mission is only applied in the frozen region (i.e. mostly above the freezing level). In the tropics, this level can be roughly approximated between 4-5 km. However, the correlation coefficient does not drop importantly below those heights





(e.g. see the second row in Fig. 5). Since below the freezing height the contribution of liquid to the used integrated water content is zero, the high correlation means that the contribution to the PRO $\Delta\Phi$ in the areas that come from the frozen part of the rays is enough to reproduce the water content patterns. That is, the contribution of the liquid precipitation into $\Delta\Phi$ does not seem to have an effect on changing the pattern of global features.

Large differences appear in the lower heights in the mid-latitudes. It is of special concern the case of southern oceans (Fig. 5-
h), where the correlation coefficient drops significantly below 5 km. This means that PAZ $\Delta\Phi$ does not capture well the ice water content patterns from Cloudsat below these heights. It appears that PAZ $\Delta\Phi$ largely underestimates the water content observed by Cloudsat. Therefore, this issue seems that would not be resolved by accounting for the liquid water content part. Mixed phase clouds in the Southern Oceans (which are present in the lower heights -cloud tops reaching 3/4 km- in these areas) have posed a longstanding issue for observations (e.g. Mace et al., 2021). Hence, further work must be carried out to assess
whether the discrepancies between PAZ $\Delta\Phi$ and Cloudsat water content in these areas are due to microphysical reasons (e.g. smaller ice particles, lack of preferred orientation, etc.), observational errors, or still unaccounted factors.

Differences between observations over ocean and over land, specifically for the tropics, are further investigated and the results are shown in Figures 6 and 7. The vertical distribution of microphysical quantities such as the cloudsat reflectivity, distance within the influence of ice water content, retrieved ice water content, and particle size distribution parameters are
compared for a subset of cases where the associated infrared $Tb_{11} < 205\,\mathrm{K}$. These correspond to heavily precipitating areas, and are further split between cases over ocean and over land. The same conditions are applied on the PAZ $\Delta\Phi$ dataset, so that they can be compared. The first thing to notice is that the mean and the 85th percentile of the integrated ice water content (shown as a black and gray dashed lines on top of the histograms in Fig. 6) are similar for the cases over ocean and over land. Very small differences are noted if one focuses on the upper levels (where the mean IWC is slightly larger over land),
which correspond to small differences pointing to the same direction in the vertical distribution of the microphysical quantities. However, the mean and 85th percentile of $\Delta\Phi$ (shown as a orange solid and red dashed lines on top of the histograms in Fig. 6) are significantly different over ocean and over land. PRO $\Delta\Phi$ is larger over ocean than over land for similar amounts of integrated ice water content. This difference is likely associated to differences in the microphysics, such as different frozen particles shapes, or preferred orientations. Differences in microphysical parameters of this kind are also observed in (e.g. Gong
et al., 2018), where these are linked to the diurnal cycle variation.

In comparing PAZ with Cloudsat observations, we are comparing observations obtained at different local times. This is due to the difference in the equator crossing local times of the two satellite platforms (6AM/PM vs 9:30AM/PM). While this could induce some discrepancies in the global comparisons (with stronger effect over land), for the subset of observations chosen for the Fig. 6 study, the effect of the diurnal cycle of precipitation is expected to be minimized, since the comparison is performed
for storms that are already well developed. Therefore, the differences in microphysics (mostly orientation) must be related to the differences in physical processes taking place over ocean with respect to those happening over land. A dedicated study would be necessary to identify the causes, but a potential application of PRO in studies about microphysics is highlighted by the fact that these differences are detected.





The ratio between the Cloudsat-RO along-ray integrated ice water content and $\Delta\Phi$ aims at empirically relate both quantities.
However, several considerations must be taken into account. Unlike with the correlation coefficient, here the lack of liquid
contribution into the water content affects the result, since the ratio would be higher when liquid is considered. This would be
true for the layers below 4-5 km, specially in the tropics, and therefore, conclusions must focus on the heights above freezing
level. For these regions and where the correlation coefficient is high (i.e. bold values in Tables 1 and 2), the ratio provides an
empirical estimate of the relationship between $\Delta\Phi$ and along-ray integrated ice water content. This is important for attempting
a retrieval of such water content using PRO $\Delta\Phi$.

Single particle forward scattering simulations have been used to assess whether the relationships obtained between water
content and $\Delta\Phi$ are reliable, according to the typical types of hydrometeors found in clouds. The results shown in Fig 8 and
Table 3 corroborate that the observed ratios in the regions with high correlation coefficients fit very well within the expected
ranges. Furthermore, results obtained here seem compatible with those by Brath et al. (2020).

The robust relationships between PRO $\Delta\Phi$ and integrated ice water content has an implicit and important implication:
it demonstrates the systematic presence of horizontally oriented frozen particles thorough the different vertical cloud layers
(above freezing level), and for all latitudes (with some caveats on the lower heights for mid- and high- latitudes). This conclu-
sion expands upon results in Gong and Wu (2017); Zeng et al. (2019), where the presence of horizontally oriented particles
was observed globally using the Global Precipiation Mission Microwave Imager (GPM-GMI) polarization differences at 89
and 166 GHz. However, here we include the vertical information lacking to off-nadir looking microwave measurements.

Furthermore, the consistency and similitude of the mean, 80th and 90th percentile profiles of IWC/$\Delta\phi$ (see Fig. 5) indicate
little dispersion of the ratio values at each altitude, region and surface type (ocean/land). Therefore, $\Delta\phi$ can be seen as a robust
proxy for water content, enabling a simple way to invert the vertical profiles of polarimetric phase shift to vertical profile of
integrated water content. At the current data volume rate of 250 RO/day, and assuming a 10-year life for ROHP, a data record
of approx. 1 million ROHP observations will be available near the end of this decade. This number may be complemented by
polarimetric RO observations collected by and/or for observational weather agencies.

*Author contributions.* All co-authors have reviewed, discussed, and agreed to the final version of the manuscript. Conceptualization: RP,
EC, FJT; Data analysis: RP; Investigation: RP, EC, FJT; Writing – original draft preparation: RP; Writing – review & editing: RP, EC, FJT;
Funding acquisition: EC, FJT.

*Competing interests.* The authors declare no competing interests

*Acknowledgements.* RP has received funding from the postdoctoral fellowships program Beatriu de Pinós, funded by the Secretary of Univer-
sities and Research (Government of Catalonia) and by the Horizon 2020 program of research and innovation of the European Union under the





Marie Sklodowska-Curie grant agreement No 801370. This work was also partially supported by the program Unidad de Excelencia María de Maeztu CEX2020-001058-M. The ROHP-PAZ project is part of the Grant RTI2018-099008-B-C22 funded by the Spanish Ministry of Science and Innovation MCIN/AEI/10.13039/501100011033 and by "ERDF A way of making Europe" of the "European Union". Part of the investigations are done under the EUMETSAT ROM SAF CDOP4. FJT acknowledges support from NASA under the US Participating Investigator (USPI) program. The work performed by FJT was conducted at the Jet Propulsion Laboratory, California Institute of Technology, under a contract with NASA.



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
