# Peer review of "On the global relationship between polarimetric radio occultation differential phase shift and ice water content"

_Atmospheric Chemistry and Physics, 2022_

## Referee Comment (RC1)

**Review of "On the global relationship between polarimetric radio occultation observable . . ." by Padullés, Cardellach and Turk**

This manuscript deals with a novel use of the signal transmitted by GNSS satellites, denoted as ROHP. These signals are recorded in a limb sounding geometry, but, in contrast to standard radio occultation, polarised information is recorded and the differential phase shift between V and H is derived. As demonstrated in the manuscript, this phase shift is related to the mass, shape and orientation of ice hydrometeors in the atmosphere. Other satellite observations in combination with in situ data give us a fair knowledge of the distribution of mass, shape and sizes of ice hydrometeors, while the orientation of larger ice particles is still largely an open question. The most interesting aspect of the manuscript is then to what extent this new technology can constrain ice particle orientation. Such information is critically needed, for example, to make full use of both existing (e.g. GMI) and future (e.g. ICI) passive microwave data in numerical weather prediction (Barlakas et al., 2021).

Accordingly, the basic objective of the study is highly relevant. I said yes to review the manuscript just because we need information on orientation as input to our simulations to set up operational retrievals for ICI. Also the methodology is good. A statistical approach where CloudSat data are used to generate synthetic phase difference data is applied. A statistical comparison is needed as it is difficult to obtain collocations with CloudSat, and largely the same approach has been used to study other ice hydrometeor properties by Kulie et al. (2010) and Ekelund et al. (2020).

On the other hand, I find several weaknesses in both the analysis and the presentation. Interesting figures are presented but it is hard to judge the robustness of the results. As a consequence, I find little new solid information in the manuscript, compared to what we know from studies based on GMI. Despite this, there is hardly no review or comparison to older work. My overall judgement is that a major revision is needed to meet the standards of ACP.

Some words before going into the detailed comments. A main quantity of the manuscript is the ratio between IWC and phase difference. For simplicity, let me define this ratio as $R$:

$$R = \frac{\text{IWC}}{\Delta\Phi}.$$

Compared to GMI, the main advantage of ROHP is that vertical information can be obtained and, in my opinion, the profiles of derived $R$-values (Fig. 5) are the most interesting results. However, the accuracy of derived $R$-values can be questioned (as 2B-CWC-RO used). In any case, it would be good to have an estimation of the uncertainty of $R$.

Further, I encourage the authors to see the retrieval of $R$ as the main strength of ROHP, and not just as a step towards estimating IWC. As described below, the retrieval IWC precision could be poorer than the results seem to indicate. In addition, the poor horizontal resolution of limb sounding observations is a severe drawback if not the information is truly unique (said based on personal experience of limb IWC retrievals).

General comments:

- The CloudSat retrievals are largely taken as truth. Issues around attenuation and multiple scattering are mentioned, but there are other, more important, limitations. The ice water content (IWC) retrieved from CloudSat has significant uncertainties due to assumptions on particle size distribution (PSD) and shape.

- Having the point above in mind, the choice of using the 2B-CWC-RO product is unlucky. This is an old product. As far as I know it is still based on Austin et al. (2009). A product more actively maintained is DARDAR, with latest version described in Cazenave et al.

(2019). The advantages of DARDAR are (assuming no update of 2B-CWC-RO that I have missed):

- It is based on newer in situ data and its PSD assumptions should be more realistic.
- It incorporates Calipso and thus has a higher sensitivity to ice at high altitudes.
- It operates with a "soft spheroid" particle model. Similar models are used in Sec. 4 of the manuscript. In comparison, 2B-CWC-RO assumes spherical particles (consisting of solid ice, if my memory is correct). As this is inconsistent with the basic results of the manuscript it is a bit of contradiction to use 2B-CWC-RO.

Another difference between DARDAR and 2B-CWC-RO is the interpretation of reflectivites at temperature between -20°C and 0°C. DARDAR assumes that all back-scattering comes from ice hydrometeors, while 2B-CWC-RO assumes a gradual change from ice to liquid. This difference and the incorporation of Calipso in DARDAR should have a significant impact on $R$-values obtained.

That is, including DARDAR would give higher confidence in the results. To be clear, to reflect the uncertainty in the reference data, both 2B-CWC-RO and DARDAR should be included (at least when it comes to mean values).

- The manuscript shows mean profiles of $R$. Some differences between land and ocean are noted and discussed. In line with studies based on GMI, the mean polarisation signal appears to be relatively stable and varies little with e.g. latitude (Gong and Wu, 2017). However, it must be remembered that these are mean values, and they do not imply that the same $R$ is valid for individual observations. There could be large local variations in $R$ but the mean value could still be relatively constant. In fact, in Kaur et al. (2022) we show that GMI observations only can be understood by a distribution of shapes and orientations, resulting in cases giving different degree of polarisation. This matches a distribution of $R$. It would be interesting if the authors could find a way to estimate the variation of $R$, around its mean.

  As far as I noted, this aspect is not considered in the manuscript, but has important consequences. Most importantly, this means that the relationship between a single $\Delta\Phi$ and IWC could in fact be weak. That the robust relationship is just valid for averages. The authors suggest the ROHP as a way to measure IWC, but there is no discussion of the impact of this issue on the IWC retrieval precision. To be clear, hail and cases where the particles exhibit totally random orientation should give very small $\Delta\Phi$, despite substantial IWC.

  I still see a value in more ROHP measurements, but not as a way to measure IWC. The selling point for ROHP, I consider to be the unique information on shape/orientation.

- In line with the last point, there could exist situations with $\Delta\Phi = 0$, but IWC $> 0$. This combination leads to $R = \infty$. That is, it would be better to define $R$ as $\Delta\Phi/$IWC. It also feels more natural that spherical particles result in a factor that is 0 (such as $\rho$ introduced in Barlakas et al. (2021).

- There should be some noise in the measurement of $\Delta\Phi$. To what extent is derived $R$ affected by this noise? How are negative $R$ values treated? Any measurements giving a negative $R$ well above the noise level?

- Older studies using passive microwave data exploring the polarisation signatures of oriented particles are poorly reflected in the manuscript. Gong and Wu (2017), Gong et al. (2018) and Zeng et al. (2019) are mentioned, but there is no real discussion if the findings are consistent or not with these older papers. Other older studies to consider include Defer et al. (2014) and Kaur et al. (2022).

- The reader could get the impression that these are the first limb sounding measurements of ice hydrometeors, and the work on passive microwave limb sounding should be acknowledged. In particular the work on Aura MLS by Dong Wu, e.g. Wu et al. (2009). There is in fact even a study based on Aura MLS looking into shape/orientation using polarisation (Davis et al., 2005), and the results of that study should be considered.

- I don't doubt that there could be differences in microphysics between ocean and land areas, but I don't find the analysis performed in Sec. 3.3 sufficient to rule out diurnal variations as the cause to the deviating results obtained for land. (That a cold 11 $\mu$m radiance is found somewhere in the neighbourhood does not guarantee an apple-to-apple comparison. For example, the convective systems can still be either in an early or late stage.) Limb sounding results could again be used reference. In fact, CloudSat and observations at 6:00/18:00 are combined in Eriksson et al. (2010), exactly as here for ROHP, and large differences in IWC over land due to the local time sampling are shown. Further data on diurnal variations of IWC are found in Eriksson et al. (2014).

- The data presented in Sec. 4 are good and interesting, but need a better presentation. Most importantly, there are too many colours and symbols in Fig. 8 to safely discern the lines. The Abstract says ".. horizontally oriented aggregated ice particles and tilted pristine ice plates agree well with the observations". And I don't see how this claim is backed up. Anyhow, I don't think it makes sense to pick out a single particle model as best, considering the variations in mean $R$ and the uncertainties discussed above. Accordingly, the most important part of Sec. 4 is to derive the general tendencies, such as how $R$ varies with effective density, axis ratio and wetness. But I don't find any clear statements (or a figure) clarifying this.

- There are several comments around the impact of liquid particles that need to be clearer. For example, the text on line 236 indicates that the authors thinks that liquid particles contribute significantly to $\Delta\Phi$, while line 363 indicates the opposite. Some quantitative values would be good. Have the authors in any way estimated the possible $\Delta\Phi$ induced by liquid drops?

- There should be a proper Conclusion section. The present Sec. 5 makes it hard to extract the main outcomes.

Some minor comments:

- Without an explanation term $\Delta\Phi$ is understood by very few persons, and the title should be changed (with $\Delta\Phi$ explained in words).

- The first paragraphs of the Introduction is hard to follow. Anyhow, I don't find this and the second paragraph relevant for the main results of the study (nor to match the manuscript's title).

- Equations should be expressed in terms of SI units, to avoid confusion and the need to state units. (Another unit can still be used in figures. For example, no problem to plot IWC in g/m$^3$.)

- Line 122: No need to inform the reader what units that are used internally.

- Line 139: Huang et al (2015) is cited as reference for typical IWP values. I had a looked in the reference for curiosity and I must say that the values look to be far too low. According to Fig. 5 in Huang et al (2015) mean IWP in the tropics is about $1\,\mathrm{g/m}^2$. This about two orders of magnitude lower than what DARDAR reports, see e.g. Duncan and Eriksson (2018); Kaur et al. (2022).

- Figure 3: Any explanation for the "spike" over land around -60 degrees?

- Figure 5: Place altitude on y-axis, as done in Fig. 6.

- Line 404: I agree that the results appear compatible with Brath et al (2020), but I assume the general reader would need an explanation.

Kind regards, Patrick Eriksson

**References**

Austin, R. T., Heymsfield, A. J., and Stephens, G. L. (2009). Retrieval of ice cloud microphysical parameters using the cloudsat millimeter-wave radar and temperature. *J. Geophys. Res.*, 114(D8).

Barlakas, V., Geer, A. J., and Eriksson, P. (2021). Introducing hydrometeor orientation into all-sky microwave and submillimeter assimilation. *Atmos. Meas. Tech.*, 14(5):3427–3447.

Cazenave, Q., Ceccaldi, M., Delanoë, J., Pelon, J., Groß, S., and Heymsfield, A. (2019). Evolution of dardar-cloud ice cloud retrievals: new parameters and impacts on the retrieved microphysical properties. *Atmos. Meas. Tech.*, 12(5):2819–2835.

Davis, C., Wu, D., Emde, C., Jiang, J., Cofield, R., and Harwood, R. (2005). Cirrus induced polarization in 122 GHz Aura Microwave Limb Sounder radiances. *Geophys. Res. Lett.*, 32(14).

Defer, E., Galligani, V. S., Prigent, C., and Jimenez, C. (2014). First observations of polarized scattering over ice clouds at close-to-millimeter wavelengths (157 ghz) with madras on board the megha-tropiques mission. *J. Geophys. Res.*, 119(21):12–301.

Duncan, D. I. and Eriksson, P. (2018). An update on global atmospheric ice estimates from satellite observations and reanalyses. *Atmos. Chem. Phys.*, 18(15):11205–11219.

Ekelund, R., Eriksson, P., and Pfreundschuh, S. (2020). Using passive and active observations at microwave and sub-millimetre wavelengths to constrain ice particle models. *Atmos. Meas. Tech.*, 13(2):501–520.

Eriksson, P., Rydberg, B., Johnston, M., Murtagh, D. P., Struthers, H., Ferrachat, S., and Lohmann, U. (2010). Diurnal variations of humidity and ice water content in the tropical upper troposphere. *Atmos. Chem. Phys.*, 10(23):11519–11533.

Eriksson, P., Rydberg, B., Sagawa, H., Johnston, M. S., and Kasai, Y. (2014). Overview and sample applications of SMILES and Odin-SMR retrievals of upper tropospheric humidity and cloud ice mass. *Atmos. Chem. Phys.*, 14(23):12613–12629.

Gong, J. and Wu, D. L. (2017). Microphysical properties of frozen particles inferred from global precipitation measurement (gpm) microwave imager (gmi) polarimetric measurements. *Atmos. Chem. Phys.*, 17(4):2741–2757.

Gong, J., Zeng, X., Wu, D. L., and Li, X. (2018). Diurnal variation of tropical ice cloud microphysics: Evidence from global precipitation measurement microwave imager polarimetric measurements. *Geophys. Res. Lett.*, 45(2):1185–1193.

Kaur, I., Eriksson, P., Barlakas, V., Pfreundschuh, S., and Fox, S. (2022). Fast radiative transfer approximating ice hydrometeor orientation and its implication on IWP retrievals. *Remote sensing*, 14(7).

Kulie, M. S., Bennartz, R., Greenwald, T. J., Chen, Y., and Weng, F. (2010). Uncertainties in microwave properties of frozen precipitation: Implications for remote sensing and data assimilation. *J. Atmos. Sci.*, 67(11):3471–3487.

Wu, D., Austin, R., Deng, M., Durden, S., Heymsfield, A., Jiang, J., Lambert, A., Li, J.-L., Livesey, N., McFarquhar, G., et al. (2009). Comparisons of global cloud ice from mls, cloudsat, and correlative data sets. *Journal of Geophysical Research: Atmospheres*, 114(D8).

Zeng, X., Skofronick-Jackson, G., Tian, L., Emory, A. E., Olson, W. S., and Kroodsma, R. A. (2019). Analysis of the global microwave polarization data of clouds. *Journal of Climate*, 32(1):3–13.

---

## Author Comment (AC1)

**Review of "On the global relationship between polarimetric radio occultation observable . . . "
by Padullés, Cardellach and Turk**

Dear Patrick,

First of all, we would really like to thank you for the extensive review you provided. We understand the amount of work and time you dedicated to it, and we really appreciate it.

We mostly agree with your comments and suggestions. We have done a lot of work to improve the manuscript, both in the analysis and the presentation. Below there are point-by-point responses to all the reviewer's comments. In general terms, there are two main things we have done:
(1) We have repeated the analysis using the DARDAR product. This changed the results especially near the freezing level. The changes have not been dramatic, but the discussion part has been re-written accordingly. Furthermore, we have evaluated and discussed the uncertainties for R.
(2) We have changed the way we presented the results in Section 4. Different analysis and different plots are shown now, and we believe that our points are made clearer.

Below you will find some comments for each point you raised. Again, thanks for your time reviewing this article.

Ramon Padullés, on behalf of the authors.

This manuscript deals with a novel use of the signal transmitted by GNSS satellites, denoted as ROHP. These signals are recorded in a limb sounding geometry, but, in contrast to standard radio occultation, polarised information is recorded and the differential phase shift between V and H is derived. As demonstrated in the manuscript, this phase shift is related to the mass, shape and orientation of ice hydrometeors in the atmosphere. Other satellite observations in combination with in situ data give us a fair knowledge of the distribution of mass, shape and sizes of ice hydrometeors, while the orientation of larger ice particles is still largely an open question. The most interesting aspect of the manuscript is then to what extent this new technology can constrain ice particle orientation. Such information is critically needed, for example, to make full use of both existing (e.g. GMI) and future (e.g. ICI) passive microwave data in numerical weather prediction (Barlakas et al., 2021).

Accordingly, the basic objective of the study is highly relevant. I said yes to review the manuscript just because we need information on orientation as input to our simulations to set up operational retrievals for ICI. Also the methodology is good. A statistical approach where CloudSat data are used to generate synthetic phase difference data is applied. A statistical comparison is needed as it is difficult to obtain collocations with CloudSat, and largely the same approach has been used to study other ice hydrometeor properties by Kulie et al. (2010) and Ekelund et al. (2020).

On the other hand, I find several weaknesses in both the analysis and the presentation. Interesting figures are presented but it is hard to judge the robustness of the results. As a consequence, I find little new solid information in the manuscript, compared to what we know from studies based on GMI. Despite this, there is hardly no review or comparison to older work. My overall judgement is that a major revision is needed to meet the standards of ACP.

Some words before going into the detailed comments. A main quantity of the manuscript is the ratio between IWC and phase difference. For simplicity, let me define this ratio as R:

$R = IWC / \Delta\Phi$

Compared to GMI, the main advantage of ROHP is that vertical information can be obtained and, in my opinion, the profiles of derived R-values (Fig. 5) are the most interesting results. However, the accuracy of derived R-values can be questioned (as 2B-CWC-RO used). In any case, it would be good to have an estimation of the uncertainty of R. Further, I encourage the authors to see the retrieval of R as the main strength of ROHP, and not just as a step towards estimating IWC. As described below, the retrieval IWC precision could be poorer than the results seem to indicate. In addition, the poor horizontal resolution of limb sounding observations is a severe drawback if not the information is truly unique (said based on personal experience of limb IWC retrievals). General comments:

- The CloudSat retrievals are largely taken as truth. Issues around attenuation and multiple scattering are mentioned, but there are other, more important, limitations. The ice water content (IWC) retrieved from CloudSat has significant uncertainties due to assumptions on particle size distribution (PSD) and shape.
- Having the point above in mind, the choice of using the 2B-CWC-RO product is unlucky. This is an old product. As far as I know it is still based on Austin et al. (2009). A product more actively maintained is DARDAR, with latest version described in Cazenave et al. (2019). The advantages of DARDAR are (assuming no update of 2B-CWC-RO that I have missed):
  - It is based on newer in situ data and its PSD assumptions should be more realistic.
  - It incorporates Calipso and thus has a higher sensitivity to ice at high altitudes.
  - It operates with a "soft spheroid" particle model. Similar models are used in Sec. 4 of the manuscript. In comparison, 2B-CWC-RO assumes spherical particles (consisting of solid ice, if my memory is correct). As this is inconsistent with the basic results of the manuscript it is a bit of contradiction to use 2B-CWC-RO.

  Another difference between DARDAR and 2B-CWC-RO is the interpretation of reflectivites at temperature between -20∘C and 0∘C. DARDAR assumes that all back-scattering comes from ice hydrometeors, while 2B-CWC-RO assumes a gradual change from ice to liquid. This difference and the incorporation of Calipso in DARDAR should have a significant impact on R-values obtained. That is, including DARDAR would give higher confidence in the results. To be clear, to reflect the uncertainty in the reference data, both 2B-CWC-RO and DARDAR should be included (at least when it comes to mean values).

  We have repeated the whole analysis using the IWC retrievals from DARDAR V3. The results have changed, especially in the height layers near the freezing level. The results using 2B-CWC-RO have not been kept, but a simple comparison between the integrated IWC using the two products is provided in the analysis.
  We believe, however, that to go beyond that (e.g. an analysis of the uncertainties in the different IWC datasets, etc.) is way out of the scope of this work. We emphasize in the discussion that the results depend on the IWC retrieval that is being used, and briefly comment the differences.

- The manuscript shows mean profiles of R. Some differences between land and ocean are noted and discussed. In line with studies based on GMI, the mean polarisation signal appears to be relatively stable and varies little with e.g. latitude (Gong and Wu, 2017). However, it must be remembered that these are mean values, and they do not imply that the same R is valid for individual observations. There could be large local variations in R but the mean value could still be relatively constant. In fact, in Kaur et al. (2022) we show that GMI observations only can be understood by a distribution of shapes and orientations, resulting in

cases giving different degree of polarisation. This matches a distribution of R. It would be interesting if the authors could find a way to estimate the variation of R, around its mean.

As far as I noted, this aspect is not considered in the manuscript, but has important consequences. Most importantly, this means that the relationship between a single $\Delta\Phi$ and IWC could in fact be weak. That the robust relationship is just valid for averages. The authors suggest the ROHP as a way to measure IWC, but there is no discussion of the impact of this issue on the IWC retrieval precision. To be clear, hail and cases where the particles exhibit totally random orientation should give very small $\Delta\Phi$, despite substantial IWC.

I still see a value in more ROHP measurements, but not as a way to measure IWC. The selling point for ROHP, I consider to be the unique information on shape/orientation.

We have emphasized the points made by the reviewer in the text. First of all, in this paper we are not attempting any retrieval of IWC, but we only mentioned it as a potential way forward – something we have removed. We emphasize as well that the robust relationship holds for mean values, but that we could find cases with large IWC yielding small $\Delta\Phi$ (and we cite Gond and Wu, 2017, as an example of very cold Tb with low PD). Not having Cloudsat-PAZ coincident measurements pose a challenge in assessing this, but we are currently thinking and conducting additional studies along this line.
The distribution of R was taken into account in the previous version of the paper with the 80[th] and 90[th] percentiles. The correlation coefficient between the higher percentiles, and its ratios, quantify how well the distributions of Rs behave at higher ends. In this revised version, we keep the higher percentiles in the analysis, along with the mean values, and we include a measure of the uncertainty of R around its mean (i.e. one standard deviation), obtained from a linear fit between $\Delta\Phi$ and IWC. To include this, we have separated the old Figure 5 in two plots, one for the correlation coefficient and one for the ratios.

- In line with the last point, there could exist situations with $\Delta\Phi = 0$, but IWC > 0. This combination leads to R = $\infty$. That is, it would be better to define R as $\Delta\Phi$/IWC. It also feels more natural that spherical particles result in a factor that is 0 (such as $\rho$ introduced in Barlakas et al. (2021).
  We followed reviewers suggestion on that, and in this revised version of the paper, the ratio is defined as $\Delta\Phi$/IWC.

- There should be some noise in the measurement of $\Delta\Phi$. To what extent is derived R affected by this noise? How are negative R values treated? Any measurements giving a negative R well above the noise level?
  Yes, there is some noise. Noise for single $\Delta\Phi$ measurements is not taken into account since the propagation of such noise when computing the mean values for the climatology disappears. However, the effect of the noise can be seen around 0 IWC in, for example, Figure 4. The noise is therefore included in the dispersion affecting the mean R value.

- Older studies using passive microwave data exploring the polarisation signatures of oriented particles are poorly reflected in the manuscript. Gong and Wu (2017), Gong et al. (2018) and Zeng et al. (2019) are mentioned, but there is no real discussion if the findings are consistent or not with these older papers. Other older studies to consider include Defer et al. (2014) and Kaur et al. (2022).
  We have included a few sentences about these results in the discussion.

- The reader could get the impression that these are the first limb sounding measurements of ice hydrometeors, and the work on passive microwave limb sounding should be acknowledged. In particular the work on Aura MLS by Dong Wu, e.g. Wu et al. (2009). There is in fact even a study based on Aura MLS looking into shape/orientation using polarisation (Davis et al., 2005), and the results of that study should be considered.
  We did not mention this work here because we already did in the first paper related to this topic, Padulles et al. 2022, and we did not want to copy the same here. However, we have found a way to discuss the work on MLS in the introduction.

- I don't doubt that there could be differences in microphysics between ocean and land areas, but I don't find the analysis performed in Sec. 3.3 sufficient to rule out diurnal variations as the cause to the deviating results obtained for land. (That a cold 11 µm radiance is found somewhere in the neighbourhood does not guarantee an apple-to-apple comparison. For example, the convective systems can still be either in an early or late stage.) Limb sounding results could again be used reference. In fact, CloudSat and observations at 6:00/18:00 are combined in Eriksson et al. (2010), exactly as here for ROHP, and large differences in IWC over land due to the local time sampling are shown. Further data on diurnal variations of IWC are found in Eriksson et al. (2014).
  After considering this reviewer's suggestion, we have decided to drop this part of the analysis. To properly evaluate the effect of the diurnal cycle would require an amount of work that may be worth an additional paper. We mention the possibility that diurnal cycle differences in the observations yield some differences, especially over land, and we leave it for future work.

- The data presented in Sec. 4 are good and interesting, but need a better presentation. Most importantly, there are too many colours and symbols in Fig. 8 to safely discern the lines. The Abstract says ".. horizontally oriented aggregated ice particles and tilted pristine ice plates agree well with the observations". And I don't see how this claim is backed up. Anyhow, I don't think it makes sense to pick out a single particle model as best, considering the variations in mean R and the uncertainties discussed above. Accordingly, the most important part of Sec. 4 is to derive the general tendencies, such as how R varies with effective density, axis ratio and wetness. But I don't find any clear statements (or a figure) clarifying this.
  We have completely re-shaped Section 4. We believe that now it is simpler and makes the points we wanted to make clearer. New Figure 7 shows the results of Kdp vs IWC when changing the different parameters we can play with in the simulations. It shows how several combinations of effective density, axis ratio, and orientation of particles can yield similar results. Furthermore, a simple study mimicking the approach followed in the previous Sections of the paper relates the simulated $\Delta\Phi$ with the IWC, and finds the best match comparing to the results obtained in Section 3.

- There are several comments around the impact of liquid particles that need to be clearer. For example, the text on line 236 indicates that the authors thinks that liquid particles contribute significantly to $\Delta\Phi$, while line 363 indicates the opposite. Some quantitative values would be good. Have the authors in any way estimated the possible $\Delta\Phi$ induced by liquid drops?
  Since in this paper we focus on the relationship with IWC, we have decided to truncate all observed and simulated profiles at the freezing level. This way, the contribution of liquid phase precipitation is minimized.

- There should be a proper Conclusion section. The present Sec. 5 makes it hard to extract the main outcomes.

Even though we have re-shaped Sect. 5, we have also included a short conclusions section stating the main points and outcomes of the study.

**Some minor comments:**

- Without an explanation term ΔΦ is understood by very few persons, and the title should be changed (with ΔΦ explained in words).
  Now the title reads: On the global relationship between polarimetric radio occultation differential phase shift and ice water content

- The first paragraphs of the Introduction is hard to follow. Anyhow, I don't find this and the second paragraph relevant for the main results of the study (nor to match the manuscript's title).
  We have removed the two first paragraphs.

- Equations should be expressed in terms of SI units, to avoid confusion and the need to state units. (Another unit can still be used in figures. For example, no problem to plot IWC in g/m3 .)
  We have modified Eq. 3 accordingly. However, we left Eq. 2 as it was for two reasons: it is consistent with what we have already written in previous Polarimetri RO papers, and because it emphasizes the fact that ΔΦ is measured in mm. This is an important point because readers from the polarimetric radar community could get confused if not clearly stated.

- Line 122: No need to inform the reader what units that are used internally
  Ok. We have changed to text accordingly.

- Line 139: Huang et al (2015) is cited as reference for typical IWP values. I had a looked in the reference for curiosity and I must say that the values look to be far too low. According to Fig. 5 in Huang et al (2015) mean IWP in the tropics is about 1 g/m2 . This about two orders of magnitude lower than what DARDAR reports, see e.g. Duncan and Eriksson (2018); Kaur et al. (2022).
  Ok. We have removed this reference, since it was not relevant for the results of the study.

- Figure 3: Any explanation for the "spike" over land around -60 degrees?
  It was due to the lack of observations over-land around -60 degrees. A single point made the mean spike. This has been corrected.

- Figure 5: Place altitude on y-axis, as done in Fig. 6.
  Done.

- Line 404: I agree that the results appear compatible with Brath et al (2020), but I assume the general reader would need an explanation.
  Done.

Kind regards, Patrick Eriksson

**References**

Austin, R. T., Heymsfield, A. J., and Stephens, G. L. (2009). Retrieval of ice cloud microphysical parameters using the cloudsat millimeter-wave radar and temperature. J. Geophys. Res., 114(D8).

Barlakas, V., Geer, A. J., and Eriksson, P. (2021). Introducing hydrometeor orientation into all-sky microwave and submillimeter assimilation. Atmos. Meas. Tech., 14(5):3427–3447.

Cazenave, Q., Ceccaldi, M., Delanoë, J., Pelon, J., Groß, S., and Heymsfield, A. (2019). Evolution of dardar-cloud ice cloud retrievals: new parameters and impacts on the retrieved microphysical properties. Atmos. Meas. Tech., 12(5):2819–2835.

Davis, C., Wu, D., Emde, C., Jiang, J., Cofield, R., and Harwood, R. (2005). Cirrus induced polarization in 122 GHz Aura Microwave Limb Sounder radiances. Geophys. Res. Lett., 32(14).

Defer, E., Galligani, V. S., Prigent, C., and Jimenez, C. (2014). First observations of polarized scattering over ice clouds at close-to-millimeter wavelengths (157 ghz) with madras on board the megha-tropiques mission. J. Geophys. Res., 119(21):12–301.

Duncan, D. I. and Eriksson, P. (2018). An update on global atmospheric ice estimates from satellite observations and reanalyses. Atmos. Chem. Phys., 18(15):11205–11219.

Ekelund, R., Eriksson, P., and Pfreundschuh, S. (2020). Using passive and active observations at microwave and sub-millimetre wavelengths to constrain ice particle models. Atmos. Meas. Tech., 13(2):501–520.

Eriksson, P., Rydberg, B., Johnston, M., Murtagh, D. P., Struthers, H., Ferrachat, S., and Lohmann, U. (2010). Diurnal variations of humidity and ice water content in the tropical upper troposphere. Atmos. Chem. Phys., 10(23):11519–11533.

Eriksson, P., Rydberg, B., Sagawa, H., Johnston, M. S., and Kasai, Y. (2014). Overview and sample applications of SMILES and Odin-SMR retrievals of upper tropospheric humidity and cloud ice mass. Atmos. Chem. Phys., 14(23):12613–12629.

Gong, J. and Wu, D. L. (2017). Microphysical properties of frozen particles inferred from global precipitation measurement (gpm) microwave imager (gmi) polarimetric measurements. Atmos. Chem. Phys., 17(4):2741–2757.

Gong, J., Zeng, X., Wu, D. L., and Li, X. (2018). Diurnal variation of tropical ice cloud microphysics: Evidence from global precipitation measurement microwave imager polarimetric measurements. Geophys. Res. Lett., 45(2):1185–1193.

Kaur, I., Eriksson, P., Barlakas, V., Pfreundschuh, S., and Fox, S. (2022). Fast radiative transfer approximating ice hydrometeor orientation and its implication on IWP retrievals. Remote sensing, 14(7).

Kulie, M. S., Bennartz, R., Greenwald, T. J., Chen, Y., and Weng, F. (2010). Uncertainties in microwave properties of frozen precipitation: Implications for remote sensing and data assimilation. J. Atmos. Sci., 67(11):3471–3487.

Wu, D., Austin, R., Deng, M., Durden, S., Heymsfield, A., Jiang, J., Lambert, A., Li, J.-L., Livesey, N., McFarquhar, G., et al. (2009). Comparisons of global cloud ice from mls, cloudsat, and correlative data sets. Journal of Geophysical Research: Atmospheres, 114(D8).

Zeng, X., Skofronick-Jackson, G., Tian, L., Emory, A. E., Olson, W. S., and Kroodsma, R. A. (2019). Analysis of the global microwave polarization data of clouds. Journal of Climate, 32(1):3–13.

---

## Author Comment (AC2)

**Review of: On the global relationship between polarimetric radio occultation observable delta_phi and ice water content by Ramon Padulles, Estel Cardellach, and F. Joseph Turk**

Dear reviewer,

First of all, thank you very much for your time spent reviewing this manuscript. The comments and suggestions clearly contributed to improve the paper.

Following yours and reviewer #1 comments and suggestions, we have performed quite a lot of work which can be, in general, summarized as follows:

(1) We have repeated the analysis using the DARDAR product. This is a more up-to-date and well maintained product containing IWC retrievals from Cloudsat. Its algorithm performs different assumptions regarding the particle size distribution and shapes of the particles. This changed the results especially near the freezing level. The changes have not been dramatic, but the discussion part has been re-written accordingly.

(2) We have changed the way we presented the results in Section 4. Different analysis and different plots are shown now, and we believe that our points are made clearer.

Below there is a point-by-point response to all reviewer's comments.

Thanks again for reviewing this manuscript.

Ramon Padullés, on behalf of the authors.

Summary: This manuscript details research comparing the measurement of polarimetric radio occultation data to retrievals of ice water content from Cloudsat radar. The first portion compares climatology of observed RO delta_phi to Cloudsat ice water content (IWC) retrievals that have been mapped onto RO sampling geometry. The latter dataset is collated for a large sample of Cloudsat data, facilitating comparison with RO delta_phi. Following this section is an comparison between forward-simulated delta_phi and IWC based on size distributions of plausible particles related to Cloudsat IWC retrievals. Overall the research presented seems valuable. There are numerous minor wordsmithing, grammar, and typo corrections that should be made. Also, more discussion should be offered on the limitations of the Cloudsat retrievals that are treated here as a benchmark. These products involve numerous assumptions are only partly supported by the state of knowledge on the global distribution of ice properties and size distribution characteristics. The authors should try to frame the scope of the work more clearly in light of the uncertainties in Cloudsat retrievals, as well as other uncertainties related to ice phase clouds. For example, the word "verification" is excessively strong for the current work, which is closer to cross-comparison. There are a small number of  major comments (see below) that involve statements or assertions that are questionable, misleading, or just plain wrong. These should be revised. I recommend major revisions.

 Recommendation: Major revisions

**General Comments:**

The use of Cloudsat radar retrievals as a point of comparison is highly questionable and should be treated with skepticism. A sinlge W-band measurement of cloud properties is insufficient to provide a reliable estimate of the likely degrees of freedom in ice particle distributions, in particular as those particles become larger and attenuation and resonance scattering effects dominate over the small-particle-assumption ("Rayleigh") limit. I would expect in many cases that Cloudsat retrieval errors

contribute as much, if not more, to the mismatch between Cloudsat and PAZ PRO. The reasons for Cloudsat retrieval errors should be obvious, but of course includes uncertainties related to size distribution and particle property assumptions. A more robust approach would be to include ground-based radar KDP, or ground-validation campaign data that includes a comprehensive suite of instruments (for example, radar, lidar, in situ cloud probes, etc). While the scope of the current work is sufficient, that scope and its limitations should be accurately conveyed to the reader.

We agree with the reviewer. And we also believe that a comparison with radar observations would be nice. In fact, we are currently working on this, but the amount of work and time make it no feasible to be included in this analysis. We believe the two studies are complementary, and the new analysis will use many of the results presented here. Another major challenge is the amount of coincident measurements between ground-based or space-based radars, but this is being overcame as PAZ satellite keeps collecting observations.

**Major Comments:**

l56: "water content" and "ice water content" need to be made clear, given the strong differences in scattering between liquid and ice particles. Perhaps avoid "WC" should be avoided altogether, unless total water content is being shown. Instead, replace with the unambiguous "liquid water content LWC" and "ice water content IWC". For example, Fig. 2 shows ice water content, but the plots are labeled "WC". This is confusing.

We agree. We have changed all figures to show IWC. Also, now there should be no confusion since we have also masked out the non-frozen part of the observations (see answer to comment regarding l236 below).

l60: It should be mentioned here that this is performed using ground-based polarimetric radars at S-band (maybe some at C- or X-?). It is not made clear anywhere in the manuscript what frequency the PAZ operates at. This may be common knowledge to many, but should be mentioned here for completeness. The reader should not be forced, as I was, to look up that it's somewhere in the L-band (1-1.5 GHz).

We have noted that previous studies used radar observations at $S - K$ bands, and we have also included the frequency at which GPS operates.

l151: The statement that KDP and IWC "depend" on the third moment is disingenuous. It's accurate to say they are both affected by M3, but neither is likely to be proportional to it for ice or mixed-phase particles (or even liquid). One can expect a correlation, but not a unique "relationship". It's not entirely clear what you mean to suggest by "relationship", but in any case, this discussion is highly misleading and must be revised.

We have seen in the literature that some authors relate IWC and Kdp using linear relationships (e.g. Bringi and Chandrasekhar, 2001, Eq. 7.101; Nguyen et al. 2019). It is true, however, that different types of hydrometeors may have different relationships, and we believe that using the more conservative statement "affected" is more accurate.

l159: Some effort should be made to convey how you focus exclusively on glaciated regions, and avoid precipitating liquid or mixed-phase regions. Uncertainties and limitations associated approach should be discussed. An explanation of your investigation of different tangent heights can be then related to this. Why are 7km and 9km chosen, for example? Why does fig 4b not include 7km? Why are values reported in Fig 5. below 5km (where significant liquid precipitation is expected,

especially for the tropics). The authors need to do more work to support this part of their research presentation.

We have re-analyzed all data and we have truncated the profiles at the freezing level to avoid major contributions from liquid phase precipitation. It is true, though, for this study we do not account for the effect of mixed phase precipitation. This is clearly stated.

l236: Why is data below the environmental 0C level not masked?? This seems like a first-order error in your approach.

We have done this now. See previous answer.

l258: These are NOT Cloudsat "observations", they are retrievals. This is a very important point to emphasize.

We agree. Thanks. We have emphasized it in the text.

l320+: Do the authors account for the viewing angle of RO? Ie that it is not always parallel to the orientation of falling particles?

Yes, we do account for this. However, the effect is almost negligible because the angle between the incidence of the rays and the plane parallel to the Earth surface is very small even at distances far away from the tangent point (but still below 20 km, which we assume as the upper limit where we can account for any hydrometeor-related effect).

l347: There is no such proportionality. This is false.

We have used reviewers suggestion of adding "affected by the $3^{rd}$ moment" instead of saying "proportional" or "dependence".

l360-363: This discussion needs revision. The authors do not consider the possibility of, for example, compensating errors. These conclusions are a severe stretch, and must be hedged or qualified carefully.

Since we have removed the main contribution of liquid particles from the analysis, this discussion has been reformulated entirely.

**Minor Comments:**

Thanks a lot for the grammatical corrections and suggestions.

l3: replace "since that time have also" with "has also"
Done.

l3: Replace "for" with "to"
Done

l4: Replace "detection" with "detect"
Done

l8: Should be "especially"
Thanks.

l8: Remove "the" before "..major precipitation…"
Thanks.

l9: Recommend that authors hyphenate "over-ocean" and "over-land"
l10: Recommend author add "possibly" or "likely" before "involving"

l11: Replace "validated" with "evaluated" or some other such word
Done

l23 (and elsewhere): Strongly recommend that "GV" is not used for this acronym, as it is commonly used to refer to "ground validation" campaigns.
We have changed the first two paragraphs of the introduction following comments from Reviewer #1.

l25: Beginning of this sentence should be plural
Same as above.

l25: Replace "and to lower" with "and in lower"
Same as above.

l51: The reference to "it" is not clear.
Corrected.

l73: Add "us" between "enable" and "to"
Done

l81: "and has been operating until" isn't the best grammatical choice here. "has been operating" implies that it is still operating, "until" implies the opposite.
Corrected
l96: Replace "in a tangential way" with "tangentially", remove parentheses
Done

l109: Say "The first is that..." (remove "one")
Done

l131: "used" is a strange word here
We have removed it. Thanks.

l134: Reword: "Therefore, analysis of the statistics…"
Done.

l142: replace "between" with "it cannot distinguish between the effects of..." or something like that
Done

l144: "Thing" is too colloquial here, and the sentence should be reworded.
We have reworded the sentence. Thanks.

l155: Add the word "statistically" after "performed", remove "built" and "in statistical terms"
Done.

Fig. 5: Make Height the y-axis here

Done. Also, old Fig.5 has been split in two (now Fig.5 and Fig. 6).

Table 1: Is this any different data than what is in Fig. 5? Why is this a separate table???
We decided to show the results using figures and tables because we believe that the comparison with results in Section 4 are easier this way.

l221: Explain the significance of this brightness temperature.
We have removed this subsection following the suggestions and comments from Reviewer #1.

Fig. 8: It is hard to distinguish the different DDA estimates on this figure.
We agree. We have changed the plots in Section 4 and we believe that now the conclusions are more clear.

l291: It is confusing why this is referred to as a pristine ice particle, since it is unlikely that any realistic particles would form in this habit, beyond, say, frozen drops. Pristine ice particles (ie. those grown solely by vapor deposition) can have any number of densities. This statement is confusing and misleading.

We have changed the way we present and state the results in Section 4. However, in this statement we refer to the ability to simulate the forward scattering effects of all kinds of particles (from more idealized habits to aggregates and different densities / axis ratios resembling more fairly the reality). And to constrain which particles are able to reproduce reality, or not. We believe that our conclusions are now clearer.

---

## Author Response (AR2)

**Review #1**

Dear reviewer,

Thanks again to the reviewer for spending time reviewing and helping us improving the paper. We really appreciate the effort.

Following the reviewers suggestions, the major changes that the paper has undergone are the following:

- Old Figure 4 has been split in two new figures: the first two panels are now Figure 4. What was the third panel of Fig. 4 is now the first panel in Figure 5, and we have included two examples of the distribution of IWC and $\Delta\phi$ in Figure 5.

- Table 1 has been removed

- We found a bug in the way we were calculating the ratios between the p80 and p90th climatologies. Also, reading both reviewers comments we believe that such a ratio could be misleading, and we removed it from the paper. We also realized that not including these ratios only directly affected two sentences of the paper, that have been removed.

  The ratio between the mean climatologies is still included and discussed, as it was in the previous versions of the manuscript.

- In the simulations section there was a mention to DDA results. This made sense in the first version of the manuscript, but since we already removed part of these comparisons to make the analysis simpler, the mention to DDA has been removed from this version as well.

Below there is the answer to each of the comments by the reviewers.

Thanks again for the efforts reviewing this paper.

Ramon Padullés, on behalf of the authors

Thanks for following my general suggestions from the last review. I am now happy with the manuscript on the overall level, but there are still details to consider. They are listed below. I raise a number of things, but they are all relatively small and I consider this a minor revision.

- Line 8: Not sure if I agree that there is a clear difference between land and ocean. More below.

- End of abstract: The conclusion of the study is expressed quite vaguely. As I see it, the derived ratios between Kdp and IWC are the main result, and the found range should be stated (0.03-0.09 mm/km?). But also clarifying that there is uncertainty due to IWC retrievals. Again more below.

  We have modified the abstract and the conclusions to include the found range and to mention the uncertainty

- Line 20: Here (and elsewhere) you get the feeling that just "sinking" occultations are used. Is this correct? Are there not also "rising" occultations?

For the PAZ mission, only setting occultations are collected. The capability of collecting rising occultations was disabled for this particular experiment, due to the number of available ports in the receiver (in general, one port is used for setting and the other one for rising. In this case, one is used for the H antenna and the other one for the V antenna).

- Lines 21-23: This sentence can be removed. Just start next one with "The" instead of "This".

  Corrected

- Line 27: No need to bring up mm and km here. And the general rule is to use SI units.

  Corrected

- Line 31: Please clarify that "equivalent diameter" means the diameter of a sphere of solid ice having the same mass.

  Clarified

- Equation 2: The factor 1e3 is wrong. If the result should be mm/km, the factor should be 1e6. That said, it is much clearer to stick to SI units in equations (i.e. remove 1e3). This does not contradict to later still report Kdp in terms of mm/km.

  Ok, understood

- Lines 37-38: It is correct that IWC is proportional to the third moment of N(D) (with D defined as done here), but there is no general relationship between size, shape and type (and what is type?) for ice particles. Rewrite these sentences, for better clarity.

  We have removed the sentence that contained "linked to shape and type". We have also rephrased the following sentences to account for the particularities of ice water content, e.g. that it is affected by the third moment of N(D), but also other factors such as effective density, orientation, etc. must be taken into account for this work.

- Line 98: A practical question. My memory is that CloudSat orbits all start at the equator. If correct, will not your segments end up at specific latitudes, roughly 10 deg apart? For example, in Fig. 3 you report statistics for every 2 deg. How do you ensure an even sampling at 2 deg resolution?

  The number of segments each orbit is split in is lower than what would correspond to an even splitting. Then, the first segment does not correspond to the "0-index" of the cloudsat orbit, but it is initialized at a random index within what would be the first segment. From there, the segments are placed one after the other in a sequential way.

- Figure 3: Figure title says 7 km, while the text below says 8 km. What is correct? Anyhow, no need for a figure title here.

  This was an error that came from the first draft. Corrected.

- Line 157: The text says "some features are still recognizable". This indicates a very different pattern. This is not the case. For land, the black and red lines deviate a bit around the equator, but not in a dramatic way. My reaction is the opposite, that the results are surprisingly similar between land and ocean. That is, reconsider the conclusion that there is a clear difference between land and ocean.

  The feeling that there were large differences between ocean and land came from the results of the correlation coefficients in the first draft – i.e. using the CWC-RO retrieval. It is true, though, that now the results look better in terms of the ocean/land comparison. Still, the correlation coefficients are higher over tropical ocean at higher altitudes (>9km) than they are for tropical land. So, we have rephrased the conclusions to state that there are some differences (not clear differences) between ocean and land, but overall both regions show good agreement.

- Line 168: Why 80th and 90th percentiles? Why not more distinct ones? Such as, 50 and 95. Further, it would be helpful if you explain what you mean by "climatology for the 80th percentile", and that you take the ratio between the 80th percentile of Kdp and the 80th of IWC. To help the readers further, please explain what you get out of looking at these percentiles, including pointing out that if all particles have the same Kdp-IWC ratio, the ratio for the mean, 80th and 90th percentiles should all be the same. Now it is first on lines 320-321 that you make some comments in this direction.

  The choice of $80^{th}$ and $90^{th}$ percentiles is rather arbitrary, but takes into account that lower percentiles, such as $50^{th}$, could fall below the mean since the data is heavily grouped towards 0. We have included a few sentences clarifying the implications of looking at these percentiles, but the ratios between the $80^{th}$ and $90^{th}$ percentile climatologies are not shown anymore (we found a bug in their computation, and we also think that their interpretation could be misleading). Still, same conclusions apply, and minor modifications had to be done in the text.

- Sec 3: Some statements on the nature of noise are needed. How big is it for individual observations (in mm)? Normally distributed? If not, on average zero?

  We have included some statements about the noise in the third paragraph of Section 3. The noise essentially depends on the signal to noise ratio, and it is normally distributed around 0, ranging from ~2mm in the lowest layers to 0.5mm above 10km. The assessment of the noise is performed in Padulles et. al. 2020.

- Figure 6: What standard deviation is included? For me, the distinction here between standard deviation and standard error is not clear. The text speaks about a red line. There is no such line.

  What we include now is the RMSE of the linear fit we use to extract the relationship between both quantities.

- Table 1: I don't find this table very useful, are not the existing figures enough?

  Yes. This has been also pointed out by the other reviewer, so we have removed the Table.

- On the other hand, I miss a figure showing the distributions of Kdp and IWC inside a region and one altitude. That would be helpful to understand how constant Kdp/IWC is with IWC, and then also throw light on why the mean ratio is higher than the 80th and 90th percentiles ones. That is, I suggest replacing Table 1 with such a figure.

  Such a figure has been included. Old figure 4 has been splitted between the climatology maps, and the scatter-plot. The scatter plot has become Figure 5, and there are two extra panels showing the distribution of integrated IWC and PAZ Kdp for a two specific regions and heights.

- Lines 225-228: It is understandable that you don't want to go into all details here, but please be a bit more specific. The mapping of the shapes assumed for DDA to spheroids, did that follow what is described below? What do you mean by "good agreement"? Deviations of 0.1%, 1%, 10% …

  We have decided to remove this paragraph because (1) the results were not shown, and (2) the methodology applied to the comparison was actually more complex than the simplifications followed in the rest of the Section. For each shape and size the specific parameters of the equivalent spheroid were computed, something that we do not do for the general simulations – as you point out in the next comment.

- Sec 4: What you have done here is fine, but it should be clarified that the standard assumption is that the effective density decreases with particle size (i.e. not is constant as you assume). Normally expressed as m = a*Dm^b, where Dm is the maximum diameter.

  We have clarified this in the last paragraph of Section 4, and mentioned it in the conclusions.

- Line 375 and Appendix A: It's great that you compare DARDAR and 2B-CWC-RO to stress that the IWC retrievals have uncertainty. But you stop a bit early. What is the impact of the mean Kdp/IWC ratio? Is it a few %, or 50%? I suggest replacing one of the panels in Figure A1, with the ratio between the mean IWC of 2B-CWC-RO and DARDAR as a function of height. That would give values that could be used to scale the Kdp/IWC ratios found by DARDAR.

  We have followed your suggestion and we have included a panel (panel b) that shows the fractional difference between the mean DARDAR and mean 2B-CWC-RO integrated IWC as a function of height, for different latitudes. And we have included a mention to this figure at the end of Section 5, emphasizing the fact that larger discrepancies are near the freezing level while in this study we focus on the layers above.

**Review #2**

**Review of: On the global relationship between polarimetric radio occultation observable delta_phi and ice water content**

**Ramon Padulles, Estel Cardellach, and F. Joseph Turk**

**Revised version**

Dear reviewer,

Thanks again to the reviewer for spending time reviewing and helping us improving the paper. We really appreciate the effort.

Following the reviewers suggestions, the major changes that the paper has undergone are the following:

- Old Figure 4 has been split in two new figures: the first two panels are now Figure 4. What was the third panel of Fig. 4 is now the first panel in Figure 5, and we have included two examples of the distribution of IWC and $\Delta\phi$ in Figure 5.

- Table 1 has been removed

- We found a bug in the way we were calculating the ratios between the p80 and p90th climatologies. Also, reading both reviewers comments we believe that such a ratio could be misleading, and we removed it from the paper. We also realized that not including these ratios only directly affected two sentences of the paper, that have been removed.

  The ratio between the mean climatologies is still included and discussed, as it was in the previous versions of the manuscript.

- In the simulations section there was a mention to DDA results. This made sense in the first version of the manuscript, but since we already removed part of these comparisons to make the analysis simpler, the mention to DDA has been removed from this version as well.

Below there is the answer to each of the comments by the reviewers.

Thanks again for the efforts reviewing this paper.

Ramon Padullés, on behalf of the authors

......................................................................
......................................................................
Summary: This version of the manuscript is much improved from the initial submission, and better frames its results, and better details caveats and uncertainties associated with its analysis. There remain some ambiguities, awkward sections, and grammatical mistakes that need correcting. These should be straightforward to rectify in a timely fashion. I recommend minor revisions.

......................................................................
Recommendation : Minor Revisions

...........................................................
General Comments:

...........................................................
Major Comments:

- l38: WC is proportional to the third moment of a liquid DSD. However, IWC is not necessarily propotional to the third moment, as this depends on particle density. As mentioned for the first draft, IWC and WC need to be made abundantly distinct throughout the paper. More work needs to be put into that.

  We have looked carefully thorough the manuscript and made sure that Ice Water Content is used all times we refer to Water Content, specially after Section 1, where in the beginning we refer to water content in a more general way. In the first revision we already changed the word "proportional" to affected, and in this new version we emphasize the fact that for Ice there are more factors such as orientation, shape, effective density and composition that play a role in the relationship between Kdp and IWC.

- l186: I'm not clear what it means to say "the profiles have been truncated below the freezing level" --- does that mean that only RO with tangent heights above the current 0C altitude are used? Or something else? This needs to be explicitly stated somewhere. This relates importantly to figures 5 and 6, which show data in liquid and mixed-phase regions. Is the data unmasked/untruncated? Or not? It is difficult to interpret these results with these remaining methodological ambiguities.

  To truncate the profiles means that the portion of each observation whose tangent point is below freezing level is not taken into account for this study. This is why the results in the tropics below ~4 km are either non-existent or with high dispersion due to the lack of data. We have clarified this, and we also acknowledge in the end of the first subsection of Sect. 2 that for this study the effect of mixed-phase is not taken into account.

- l301: See l38 comment above

  With the clarifications made in Sect 1. (see reply to comment I38) we believe that the word "affected by" is fine.

- l357: I find this statement highly dubious. I would strongly recommend rephrasing this in terms of showing that these rations can be reproduced for different plausible distributions of ice crystals.

  We have rephrased the statement saying that the observations could potentially help constraining the plausible distributions of ice crystals that can reproduce the ratios.

...........................................................
Minor Comments:

- l17: Joint with what?

We have rephrased the sentence.

- l49: Change to "Furthermore, knowledge of XXX is crucial for..." and replace XXX with specifically what you're referring to (because its not clear right now)

Corrected

- l53: Change "its" to "their" and "their" to "MLS" to improve clarity.

Corrected

- l73: Does it still orbit? If not, change to "orbitted"

Cloudsat is actually still orbiting, but an anomaly happened in August 2020. Since August 2020 there are no available data, although as far as I know, there are plans to process some of the remaining data and make them available soon.

- l95: Change to "moving" and "rotating"

Changed

- l103: I wouldn't say "actual amount" since its really the retrieved amount

We have removed "actual"

- l140: I would remove "a lot of", which is vague and colloquial

Removed

- l146: Sentence beginning with "Being the PRO..." is a bit of a grammatical mess. Please rephrase.

We have rephrased the sentence.

- l161: Do you mean IWC or WC?

We mean IWC. We have changed it to IWC.

- l164: What does "integrated" mean here? Along the RO path? Is this explicitly spelled out anywhere?

It has been explicitly spelled in Section 2.

- l167: "integrated", "WC"

Integrated ice water content. We have changed it.

- l170: "integrated" (this time its IWC... is that the same as WC??)

We have changed all the WC to IWC.

- l209: Change to "single-particle"

Changed.

- l220: It is worth stating what ice assumptions are used for the Cloudsat IWC product, since it is used heavily throughout this work.

  We have mentioned at the end of Sect. 5 that the DARDAR V3 algorithm uses non-spherical particles but makes no assumptions about orientation, at the same time that we acknowledged that the results depend on the IWC algorithm chosen for the study.

- l269: Change to "consists of"

  Corrected

- l270: Change to "described, and proceeding.."

  Corrected

- l279-282: This is a grammatical mess and should be rephrased.

  The sentence has been rephrased.

- l301: Add "The" to beginning of sentence

  Added. Thanks for the suggestion.

- l305: Last sentence: Change to "The correlation coefficients maximize for Tropical oceans" or something

  Changed

- l309: Do you mean higher altitudes when you say "higher ends"? Please be clear.

  It refers to the higher percentiles ($80^{th}$, $90^{th}$), e.g. the tail of the distribution. This has been clarified in the text. Also, their ratios have been removed since there was a mistake in its calculation and also the way these were expressed could be misleading.

- l311: I think you should make clear that when you say this is a "longstanding issue for observations" you are referring to errors in the Cloudsat retrievals. (are you?)

  It does not refer to Cloudsat specifically, but to observations in general. In the provided reference it is stated that for these specific type of clouds present in the Southern Oceans, there is a disagreement among the observations on the amount of snow present in such clouds. We clarify that one possible explanation could relate to the retrievals withing these type of clouds.

- l315: Change to "quantifying the empirical relationship between both" or something

  Changed. Thank you for the suggestion.

- l318: I would change "accounts for non-unique relationshps" to "may be explained in part bby the non-unique…"

  Changed

- l320: Change to "The mean climatology ratio is higher for almost all…"

Changed

- l321: Change to "relates"

Changed